

# Local and Regional Enhancements of CH₄, CO, and CO₂ Inferred from TCCON Column Measurements

Kavitha Mottungan[1,a], Vanessa Brocchi[1,b], Chayan Roychoudhury[1], Benjamin Gaubert[3], Wenfu Tang[3], Mohammad Amin Mirrezaei[1], John McKinnon[1], Yafang Guo[1], Avelino F. Arellano[1,2]

[1] Department of Hydrology and Atmospheric Sciences, University of Arizona, Tucson, 85721, USA

[2] Department of Chemical and Environmental Engineering, University of Arizona, Tucson, 85721, USA

[3] NSF National Center for Atmospheric Research, Boulder, CO, 80307, USA

[a] now at: National Physical Laboratory (NPL), Teddington, UK

[b] now at: Atmo Auvergne-Rhône-Alpes, association agréé de surveillance de la qualité de l'air, 69500 Bron, France

*Correspondence to*: Avelino Arellano (afarellano@arizona.edu)

**Abstract.** In this study, we demonstrate the utility of available correlative measurements of carbon species to identify regional
and local airmass characteristics and their associated source types. In particular, we combine different regression techniques and enhancement ratio algorithms with CO, CO₂, and CH₄ data of total column abundance from 11 sites of the Total Carbon Column Observing Network (TCCON) to infer relative contributions of regional and local sources to each of these sites. The enhancement ratios provide a viable alternative to univariate measures of relationships between the trace gases that are insufficient in capturing source type and transport signatures. Regional enhancements are estimated from the difference
between bivariate regressions across a specific time window of observed total abundance of these species (BEHr) and inferred anomalies (AERr) associated with a site-specific background. Since BEHr and AERr represent the bulk and local species enhancement ratio, respectively, its difference simply represents the site-specific regional component of these ratios. We can then compare these enhancements for CO₂ and CH₄ with CO to differentiate combustion versus non-combustion associated airmasses. Our results show that while the regional and local influences in enhancements vary across sites, dominant
characteristics are found to be consistent with previous studies over these sites and with bottom-up anthropogenic and fire emission inventories. The site in Pasadena shows a dominant local influence (>60%) across all species enhancement ratios, which appear to come from a mixture of biospheric and combustion activities. In contrast, Anmyeondo shows more regionally influenced (>60%) air masses associated with high temperature and/or biofuel combustion activities. Ascension appears to only show a large regional influence (>80%) on CO/CO₂ and CO/CH₄ which is indicative of transported and combustion-
related CO from nearby African region, consistent with sharp rise in column CO (3.51±0.43 % ppb/year) in this site. These methods have important application to source analysis using space-borne column retrievals of these species.

## 1 Introduction

The rise in the abundance of greenhouse gases (e.g., CO₂, CH₄) in recent decades, because of anthropogenic activities and natural emissions associated with climate change, such as wetland, and biomass burning emissions associated with El-Niño
(Zhang et al., 2018; Kumar et al., 2023; van Vuuren and Riahi, 2008; Arneth et al., 2017), has large implications to quantifying chemistry-climate relationships. This rising trend increases the complexity in understanding the feedback mechanism (CH₄-OH-CO), retrieval bias in less validated regions or unresolved uncertainty in tropical emissions (e.g., based on TROPOspheric Monitoring Instrument (TROPOMI) and Greenhouse Gases Observing Satellite (GOSAT)) (Lunt et al., 2019; Palmer et al., 2019) and emission estimates from fossil-fuel use over growing megacities (Tang et al., 2020; Maasakkers et al., 2019).
Understanding today's regional CO₂ and CH₄ sources and sinks is a key area in carbon cycle and atmospheric composition



science given the necessity for reliable projections of future atmospheric $CO_2$ and $CH_4$ concentrations. This is especially problematic in megacities with the fastest pace of urbanization and where the anthropogenic activities are most intense, accompanied by immense energy consumption mainly in the form of fossil-fuel combustion (Kennedy et al., 2015; Grimm et al., 2008; Agudelo-Vera et al., 2012; Banerjee et al., 1999; Lamb et al., 2021). Emission estimates from fossil-fuels remain
uncertain due to poor characterization of combustion activity, efficiency and fuel-use mixtures emerging from the lack of details on pollution control strategies, energy use and combustion practices (Zhu et al., 2012; Creutzig et al., 2015; Kennedy et al., 2009; Baiocchi et al. 2015; Weisz and Steinberger, 2010; Bettencourt et al., 2007; Dodman, 2009, Bai et al., 2018). The high-efficiency combustion of fossil-fuels leads to large $CO_2$ emissions compared to CO, whereas low-efficiency combustion of residential combustion, biomass burning, among others produce more CO (Andreae and Merlet 2001; Silva and Arellano,
2017; Halliday et al., 2019; Tang et al., 2019; Wei et al., 2012; Andreae, 2019; Park et al., 2021). This uncertainty is further complicated by limited observations at the spatiotemporal scales necessary to resolve variations in combustion and fuel-use patterns (Streets et al., 2013; Nassar et al., 2013; Hutyra et al., 2014, Gately and Hutyra 2017; Creutzig et al., 2019; Arioli et al., 2020). This leads to difficulties in teasing out small anthropogenic signatures from the large natural sources and sinks dominating the carbon cycle and the uncertainties in modelling atmospheric transport (Pacala et al., 2010; Peylin et al., 2013;
Thompson et al., 2016; Erickson and Morgenstern, 2016; Oda et al., 2019; Duncan et al., 2019; Gaubert et al., 2019). This is especially true for flux estimations of $CO_2$ and $CH_4$ using top-down approaches, despite the increase in aircraft and satellite measurements of $CO_2$ and $CH_4$ abundance in recent years (Hutyra et al., 2014; Houweling et al., 2015; 2017, Chevallier, 2019; Crowell et al., 2019; Lu et al., 2021; Chandra et al., 2021). Studies have also highlighted the importance of fossil-fuel emission uncertainties on their estimates, suggesting the need for temporally defined emission inventories (Gurney et al., 2005; Peylin
et al., 2011; Thompson et al., 2016, Saeki and Patra, 2017; Gurney et al., 2020).

The abundance of a species at a particular location is mainly dependent on the variations of sources and sink. Furthermore, both regional and local transport (long-range, vertical transport and dilution in the boundary layer) influence the abundance of the species (especially in the column) and confound measurement interpretations. The major sources of $CO_2$ include anthropogenic emissions especially fossil fuel combustion, cement production, and land-use change while sinks include
uptakes by ocean and land from the atmosphere (Friedlingstein et al., 2022). While CO is primarily produced through incomplete combustion of carbon-containing fuels, oxidation of $CH_4$ and other volatile organic compounds by OH contributes to the secondary production of CO (Bakwin et al., 1995; Gaubert et al., 2016, Hoesly et al., 2018). The main chemical sink of CO in the atmosphere is OH followed by dry deposition through soil uptake (Levy 1971, Bartholomew 1981, Khalil 1990, Cordero et al 2019). This coupling of $CH_4$-OH-CO has significant impact on the growth rate and source-sink characterization
of $CH_4$ (Gaubert et al., 2017; Zhao et al., 2019; 2020; Guthrie, 1989; Prather, 1994; Lelieveld et al., 2002). Anthropogenic sources of $CH_4$ include agricultural activities (rice and livestock), solid waste, fossil fuels, and biomass burning in addition to natural sources like anaerobic ecosystems and geological activities (Saunois et al., 2020; Stavert et al., 2021). $CH_4$ and CO are thus coupled with common sources (combustion process, vehicular emission, etc.) and sink (OH) and changes in one of these



species will have a significant impact on the other (Sze, 1977; Gaubert et al., 2017). This co-variation (co-emission) or the correlations of the species can be used to derive enhancement ratios/emission ratios which vary according to source regions and source type (Suntharalingam et al., 2004; Palmer et al., 2006; Wang et al. 2010; Tang et al., 2018). For example, a recent study by Lelandais et al. (2023) uses enhancement ratios and correlations to study variability of ICOS-France observed CO, $CO_2$, and $CH_4$ in a Mediterranean climate at different regional and time scales. Their results showed 84% of their data was representative of background concentrations that were dependent on both wind speed and direction, while 16% were enhanced by anthropogenic plumes, emissions in the boundary layer, or short-term pollution events. These derived emission/enhancement ratios from multiple species are widely used to characterize emission sources (Turnbull et al., 2011, 2015; Silva et al., 2013; Anderson et al., 2014; Ammoura et al., 2014; Popa et al., 2014; Parker et al., 2016; Silva and Arellano, 2017; Bukosa et al., 2019; Tang et al., 2019; Lee et al., 2020; Sim et al., 2022; Djuricin et al., 2010) and in the flux estimation for different parts of the world (Wunch et al., 2009; Miller et al., 2012; Wennberg et al., 2012; Bozhinova et al., 2014; Super et al., 2017; Hedelius et al., 2018, Plant et al., 2022; Bares et al., 2018). For example, a recent study by Plant et al. (2022) investigated the urban emissions of $CH_4$ and CO using enhancement ratios derived from TROPOMI while Halliday et al. (2019) characterized air masses during KORUS-AQ into regions of high or low-efficiency combustion based on $CO/CO_2$ enhancement ratios derived from aircraft data. Bukosa et al. (2019) used shipborne measurements of CO, $CO_2$, and $CH_4$ to improve GHG flux estimates by comparing them with GEOS-Chem simulations to identify missing/underestimated sources in the model.

The enhancement ratio between species $X$ and $Y$ is calculated by mainly two methods: the first is by dividing the excess of $X$ by the excess of $Y$ and the second one is from a linearly regressed slope of $X$ and $Y$ (Andreae et al., 1988; Yokelson et al., 2013; Briggs, 2016). The first approach of enhancement ratio estimation requires a proper understanding of the background concentration to derive the excess abundance along with the instantaneous concentration of the species, which is not available in most cases. The ratio estimation from the regression approach has also a limitation when the emitted or locally produced species mixes with different air masses (e.g., advection from the nearby sources or mixed air masses) downwind of dominant source where measurements are made. This is especially the case for vertically integrated quantities like the column measurements (either ground-, aircraft- and satellite-based) (Cheng et al., 2017; Halliday et al., 2019; Tang et al., 2019) where vertical information of the species abundance is practically absent. If the emission or plume concentration is significantly larger than the background, the ratio from the regression slope approach does not change (Brigg et al., 2016). But, when emission of the species mixes with different 'backgrounds' than a relatively uniform field, the abundances of $X$ and $Y$ change due to mixing and/or photochemical loss (Mauzerall et al., 1998; Yokelson et al., 2013; Guyon et al., 2005); thus, making it difficult to track the locally emitted contribution to the observed abundance. Vertical and horizontal transport also complicates the interpretation of abundance and assessment of local and regional source influences at a particular location (Chatfield et al., 2020). Here, we utilize the column measurements of CO, $CO_2$, and $CH_4$ from the Total Carbon Column Observing Network (TCCON) (Wunch et al., 2011) to understand these variations in the column abundances.





The main objective of this study is to characterize the bulk characteristics of the column abundances of CO, $CO_2$, and $CH_4$ from ground based TCCON measurements using a combination of enhancement ratio approaches. Specifically, we introduce a combination of established local and bulk regression algorithms in deriving enhancement ratios of the column abundances

between these three species to understand their relationships because of emissions of these species being mixed, dispersed, transported, and transformed in the atmosphere. More importantly, we present the utility of combining these techniques in quantifying the contributions of the regional and local influences to observed columns and the corresponding enhancements observed in the respective species. We then examine the regional and seasonal variations of these influences and make use of the variability in the relationship of the multi-species enhancement ratios to infer the dominant source type leading to these

variations. While previous studies have used enhancement ratios to examine the source attribution of $CH_4$, CO, and $CO_2$ at regional and/or local scale, we note that few have investigated bulk characteristics on a source type basis using all these 3 species and using these combinations of regression algorithms for globally distributed column-integrated measurements. This proof-of-concept has an important application to on-going and planned satellite missions of these species given that TCCON measurements serve as basis for retrieval validation of these missions.

**2 Data and Methods**

**2.1 Data and Location Features**

As mentioned, we make use of the column-averaged mixing ratios of CO, $CO_2$, and $CH_4$ from the ground-based network of TCCON during the period 2012 to 2019. TCCON retrieves the column abundance from the near-infrared solar absorption spectra using high-resolution Fourier Transform Spectrometers (FTS) (Wunch et al., 2011). This network provides the column-

125 averaged dry-air mole fractions by normalizing the column abundance of the species of interest to the retrieved oxygen column abundance. The precision of the column-averaged mole fraction of $CO_2$ ($XCO_2$) is <0.25 %, $CH_4$ ($XCH_4$) is <0.3% and CO ($XCO$) is <1% under clear or partly cloudy skies (Wunch et al., 2010). TCCON data sets are widely used in the global carbon cycle studies to improve the carbon budget (source and sinks information) and for validation of atmospheric trace gas estimates retrieved from the space-based instruments such as Orbiting Carbon Observatory (OCO-2), GOSAT, GOSAT-2, and

130 TROPOMI, (Miller et al., 2007; Morino et al., 2011; Frankenberg et al., 2015; Wunch et al., 2017; Qu et al., 2021; Wang et al., 2022; Kulawik et al., 2016; Yoshida et al., 2013; Noël et al., 2022; Liang et al. 2017; Kong et al., 2019). A total of 11 TCCON sites are selected for this analysis which includes six sites in the Northern Hemispheric (NH) regions and five in the Southern Hemispheric (SH) regions and the locations are marked in Figure 1. The average column abundance retrieved at each TCCON location is embedded in the monthly spatial map of column abundances of CO from the Measurements of Pollution

In The Troposphere (MOPITT) aboard Terra, $CO_2$ from OCO-2 and GOSAT retrieved $CH_4$ during May 2018. The absence of data during May 2018 in TCCON column abundances at Darwin, Ascension, Manaus, Reunion, Hefei and Anmyeondo are shown as white circles in Figure 1. Qualitatively, MOPITT and GOSAT retrievals show reasonable agreement between the retrieval of CO, $CO_2$, and $CH_4$ column abundance relative to TCCON at these locations.



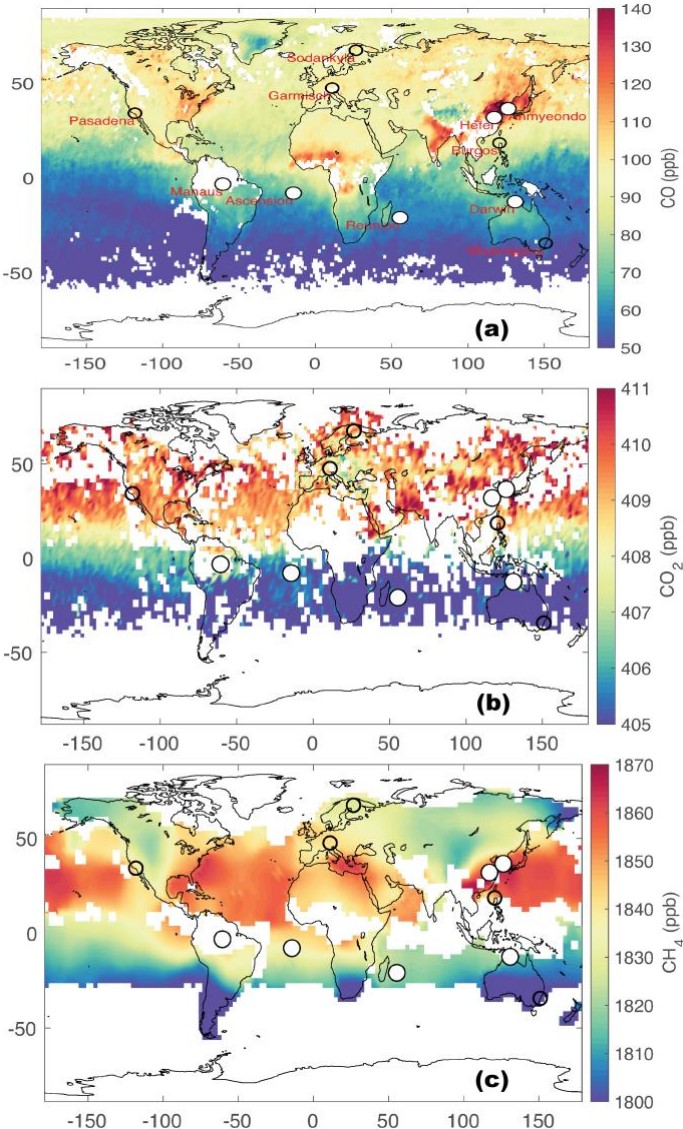

**Figure 1: May 2018 month-average abundance of: (a) CO from MOPITT, (b) CO$_2$ from OCO-2, and (c) CH$_4$ from GOSAT. Locations of TCCON sites are superimposed as black circles.**

The site in Ascension is in a small island with virtually no influence from local sources, but it captures the long-range transport of emissions from Africa (Geibel et al., 2010; Feist et al., 2014, Swap et al., 1996). Among the selected sites of study, Ascension and Reunion are representative of remote island sites located in the South Atlantic and the Indian Ocean, respectively. The humidity in the eastern part of the Reunion Island is higher than its the western counterpart. There is also a regularly occurring outflow of biomass burning emission from South Africa, Madagascar, and South America to Reunion Island (Vigouroux et al., 2012; De Maziere et al., 2017; Zhou et al., 2018). The sites in Manaus, Darwin, Garmisch, and Sodankyla are reported to



be mostly influenced by sources related to local biogenic emissions and regional anthropogenic emissions. Manaus is in the
center of the Amazon, the world's largest rainforest, and is the seventh largest city in Brazil (Dubey et al., 2014). The
measurement site in Garmisch is situated in the Alps Mountain range in Southern Germany (Sussmann and Rettinger, 2018)
while the site in Sodankyla in Northern Finland, mainly surrounded by Scots pine forest within the Fennoscandia region.
Wintertime measurements at this location is not possible due to the absence of sunlight (Kivi et al., 2022). Finally, Darwin is
the largest city in the sparsely populated Northern Territory of Australia and is situated on the Timor Sea. The site is 9 km
from the city of Darwin and adjacent to the airport (Griffith et al., 2014).

It has been previously reported that local emissions and nearby sources are significant at measurement locations in Pasadena,
Anmyeondo, and Wollongong (Griffith et al., 2014; Wennberg et al., 2015; Goo et al., 2014). The measurement site in
Pasadena is situated at the northern limit of the South Coast air basin, which is bounded by mountains on three sides and the
Pacific Ocean on the other side. The northern and eastern regions of the basin are sparsely populated deserts and receives
polluted air under normal meteorological conditions and occasionally cleaner air (Wunch et al., 2009; Wennberg et al., 2016).
In SH, the measurement site of Wollongong is representative of an urban location. The urban sources are local and is mainly
from Sydney's motorway flanks, coal mining, steelmaking facilities (Buccholz et al., 2016). Biogenic emission and bush fire
also impact the air at this site along with agricultural activities in the southwest side of the urban extent (Griffith et al., 2014;
Buchholz et al., 2016). Anmyeondo Island is located on the west coast of the Korean Peninsula, 180 km southeast of Seoul.
Although surrounding area mainly consists of agricultural lands, vegetation in and around the sites consisting of pine trees,
natural forest, and urban developments, this site is regularly influenced by Asian pollution outflows especially during Spring
(Goo et al., 2014; Oh et al., 2018).

The air in Burgos and Hefei sites are mainly dominated by regionally transported emissions (Morino et al., 2022; Liu et al.,
2022). Hefei is an inland city in the eastern part of China, and it is a rapidly developing city with a population of eight million.
The site is adjacent to a lake in flat terrain and is in the north-western rural area of Hefei city. A large anthropogenic influence
in Hefei comes mainly from heavily polluted areas in northern China and cities in the Yangtze River Delta, while natural
emissions come from cultivated lands or wetlands surrounding the site (Tian et al., 2018; Wang et al., 2017). The site in Burgos
is in a town in Ilocos Norte Province in the Philippines. This region is a coal-free province and encounters relatively clean
marine air from the western Pacific but also polluted air from long-range transport during monsoon transitions (Velazco et al.,
2017). The data period and a summary of the characteristics of these selected TCCON sites are listed in Table 1. The sites at
Pasadena, Garmisch, Reunion, Ascension, Sodankyla, Darwin, and Wollongong have longer records (> 7 years of data) as
opposed to Anmyeondo, Hefei, Manaus, and Burgos (~2 years with more gaps in between).



**Table 1: Relevant reference and acknowledgement on selected TCCON sites considered in this work.**

| Location | Data Period | Reference |
|----------|-------------|-----------|
| Pasadena | 09/2012-08/2019 | Wunch et al., 2009; Wennberg et al., 2016; Wennberg et al., 2022 |
| Ascension | 05/2012-10/2018 | Geibel et al., 2010; Feist et al., 2014 |
| Manaus | 10/2014-06/2015 | Dubey et al., 2014; Dubey et al. 2022 |
| Garmisch | 07/2007-08/2019 | Sussmann and Rettinger, 2018; Sussmann and Rettinger, 2023 |
| Sodankyla | 05/2009-06/2019 | Kivi et al., 2014; Kivi et al. 2022 |
| Anmyeondo | 02/2015-04/2018 | Goo et al., 2014; Oh et al., 2018 |
| Burgos | 03/2017-11/2018 | Velazco et al., 2017; Morino et al., 2022 |
| Hefei | 09/2015-12/2016 | Wang et al., 2017; Tian et al., 2017; Liu et al., 2023 |
| Darwin | 08/2005-09/2018 | Deutscher et al., 2014; Griffith et al., 2014; Deutscher et al., 2023 |
| Wollongong | 06/2008-11/2018 | Buchholz et al., 2016; Deutscher et al., 2023 |
| Reunion | 09/2011-02/2018 | Vigouroux et al., 2012; De Maziere et al., 2017; |
| | | Zhou et al., 2018; De Maziere et al., 2022 |

**2.2 Estimating regional and local enhancement ratios**

The observed column abundance ($C$) of any species ($spc$) retrieved at any location of TCCON measurement site ($s$) and at a particular time ($t$) is generally represented as:

$$C_{spc} = C_{true,spc} + \epsilon_{meas,spc} \tag{1}$$

where $C_{true}$ is the true species concentration being measured at $(s, t)$ and $\epsilon_{meas}$ is the measurement error. Letting $C_X = C_{CO_2}$, $C_Y = C_{CO}$, and $C_Z = C_{CO_2}$, the true concentration can be broken down into specific contributions following Levin (2003) and Turnbull (2009) as:

$$C_X = ( X_{bg} + X_{ff} + X_{bb} + X_c + X_r - X_p) + \epsilon_X \tag{2}$$

$$C_Y = ( Y_{bg} + Y_{ff} + Y_{bb} + Y_{ox} - Y_l - Y_{su}) + \epsilon_Y \tag{3}$$

$$C_Z = ( Z_{bg} + Z_{ff} + Z_{bb} + Z_{wet} + Z_{live} + Z_{oth} - Z_{cl} - Z_{su}) + \epsilon_Z \tag{4}$$

The subscripts in the above equations represent the associated sources and sinks: background ($bg$); anthropogenic processes such as fossil fuel ($ff$), biomass burning ($bb$), cement ($c$), and livestock ($live$); biospheric processes such as ecosystem respiration ($r$) and photosynthesis uptake ($p$); natural processes such as ocean ($o$), soil uptake ($su$), and wetland ($wet$); chemical processes such as oxidation from hydrocarbons ($ox$), chemical loss by OH ($l$), chemical loss by OH and Cl ($cl$); and

195 other sources ($oth$). The background component ($bg$) accounts for initial abundance, dilution, and transport processes. Direct biogenic CO emissions and oxidation of CH₄ ($Z_{cl}$) as a source of CO are included in $Y_{ox}$. We also consider the oxidation of $Y$ to $X$ as a source $X$ to be negligible in this analysis.





In this study, we adopt the following three main methods to derive enhancement ratios:

Method (1): regression of the abundances (i.e., associated linear slope from the scatter plots between $C_X$ and $C_Y$, $C_X$ and $C_Z$,
or $C_Y$ and $C_Z$). This method is denoted as Bulk Enhancement Regression Ratio (BERr) (Andreae et al., 1988; Lefer
      et al., 1994; Silva et al., 2013; Tang et al., 2019) - See Eq. 5 & 6

Method (2): ratio of $C_{spc}$ anomalies (Anomaly Enhancement Ratio or AERa) (Andreae and Merlet, 2001; Silva and Arellano,
      2017; Le Canut et al., 1996) – See Eq. 7 & 8

Method (3): regression of $C_{spc}$ anomalies (Anomaly Enhancement Regression Ratio or AERr) (Mauzerall et al., 1998;
Yokelson et al., 2013; Hobbs et al., 2003; Wunch et al., 2009; Hedelius et al., 2018; Sim et al., 2022) – See Eq. 9
      & 10

The regressions and anomaly of abundances are calculated using daily average data points across a monthly time window. The
number of daily column abundance data points available in each month at the selected TCCON location sites is provided in
Figure S1. This information is used further in the analysis for selecting the data range for comparison purposes and interpreting
the results.

**Method 1:** The enhancement ratio based on the regression of the daily average abundances of the species is considered as the
"bulk" or "global" enhancement ratio (BERr), which is interpreted to represent the sum of all the associated sources and sinks
contributions. The BERr or regression slope of daily average abundances of species $X$ and $Y$ for example is calculated simply
as the ratio of the covariance of $C_X$ and $C_Y$ to the variance of $C_X$ from a least-squares linear fit of the data. That is,

$\left(\dfrac{\Delta C_Y}{\Delta C_X}\right)_1 = \dfrac{cov\ (C_Y, C_X)}{var(C_X)}$           (5)

          $= \sum \dfrac{cov(X_{bg}, C_Y)}{var(C_X)} + \sum \dfrac{cov(X_{sources}, C_Y)}{var(C_X)} - \sum \dfrac{cov(X_{sinks}, C_Y)}{var(C_X)}$    (6)

where sources of $X = ff, bb, c, r$ and sinks $= p, o, st$ while subscript 1 denotes Method 1.

Note that for different linear regression approaches, there is a significant difference in the slope estimation when the
representation of the error ($\epsilon_{meas,spc}$) associated with the data is included (Wu and Yu, 2018). To account for the differences
in the estimates due to the choice of algorithm, we use three regression methods (Ordinary Least Square, Geometric Mean and
York) (York et al., 2004) in calculating the enhancement ratios derived based on regression approaches in Methods (1) and
(3). The enhancement ratios of BERr and AERr reported in the study are the mean of these estimates weighted by the associated
error (Verhulst et al., 2017).

**Method 2.** Local enhancement ratios are derived based on Methods (2) and (3), where the background influences/transport
components are removed from the total abundances used in Method (1) using two ways to estimate anomalies (Eq 7). That is,
1) we remove dilution/boundary layer influence from the total abundance (broadly denoted as $C_{bg,spc}$) by taking the difference
of average morning values from the average afternoon values; and 2) we remove the 'background' by calculating the difference
between the background value $C_{bg,spc}$ (assumed here as 5th percentile of the daily data) from the individual daily average
values. The anomaly of $C_{spc}$ after removing these influences from the total abundance is expressed as,





$\quad C'_{spc} = \left( C_{spc} - C_{bg,spc} \right) = \sum C_{sources} + \sum C_{sinks}$ (7)

with AERa between species $X$ and $Y$ for Method (2) for example is given by:

$$\left( \frac{\Delta C_Y}{\Delta C_X} \right)_2 = \left( \frac{C'_Y}{C'_X} \right) = \frac{\sum Y_{sources} + \sum Y_{sinks}}{\sum X_{sources} + \sum X_{sinks}}$$ (8)

**Method 3.** Accordingly, the regression slope (AERr) between species $X$ and $Y$ for Method (3) for example can be calculated

using the combination of Eqs. 5 and 7:

$\quad \left( \frac{\Delta C_Y}{\Delta C_X} \right)_3 = \frac{cov\ (C'_Y, C'_X)}{var(C'_X)}$ (9)

$$= \sum \frac{cov(X'_{sources}, C'_Y)}{var(C'_X)} - \sum \frac{cov(X'_{sinks}, C'_Y)}{var(C'_X)}$$ (10)

The regional enhancement ratio is calculated by subtracting the enhancement ratios derived based on the regression slope of

total abundances in Method (1) (BERr) from that of the ratio derived from the anomalies in Method (3) (AERr) (Cheng et al.,

2017; Briggs et al., 2016; Le Canut et al., 1996).

Similar expressions can be applied to BERr, AERa, and AERr for species $X$ and $Z$, as well as for $Y$ and $Z$.

**3 Results and Discussion**

This section describes the spatial and temporal variation (and co-variation) of $C_{spc}$ along with their corresponding local and

regional enhancement ratios. We also present in this section several qualitative inferences on the dominant processes leading

to these co-variations.

**3.1. Abundance, Trend, Seasonality, and Co-variation of CO, CO$_2$, and CH$_4$**

Firstly, it is informative to understand the spatial and temporal patterns of these species column abundance before investigating

their corresponding enhancement ratios. We show in Figure 2 the monthly variation of CO, CO$_2$, and CH$_4$ over Garmisch,

Darwin, Sodankyla, Wollongong, Pasadena, and Reunion during 2012-2019 period. TCCON sites with data gaps of shorter

time periods (see Section 2.1) are not included. The figure shows a clear seasonal cycle in the abundance of CO over all the

250 locations and the seasonal amplitude is higher over Hefei (38.3±0.0 ppb), Sodankyla (37.2±3.9 ppb) and Pasadena (36.0±4.5

ppb) compared to other locations. The seasonality in time series can indicate the presence of a non-steady state source/sink at

the location including potential regional transport into and out of the site. Furthermore, a large variability in CO is observed

in the seasonal amplitude over Burgos (15.5 ppb), Darwin (10.2 ppb), Reunion (9.2 ppb) and Wollongong (8.5 ppb) during

this period. The inter-annual variability can suggest changes in the emission sources or meteorology over time. As presented

in Table 2, the seasonal cycle of CO$_2$ and CH$_4$ is evident for TCCON sites located in the NH and those relatively closer to

emission sources such as Pasadena, Garmisch, and Sodankyla. For the other sites, the seasonal cycle appears to be low, which





can be mainly due to its remote location with relatively mixed air masses and smaller influences of local emissions (e.g., Ciais et al., 2019). The seasonal amplitude of $CO_2$ ranges from 5.6±0.0 ppm (in Hefei) to 3.6±1.9/0 ppm (in Ascension/Anemyondo). The variability in seasonal amplitude of $CO_2$ across the stations is the same (1.9 -1.2 ppm) during this period. The seasonal amplitude of $CH_4$ varies from 0.03 to 0.01 ppm across the measurement sites and its variability (~0.01 ppm) is similar in most of the locations.

The monthly mean variation of the column abundance of CO, $CO_2$, and $CH_4$ at the locations in the NH (Pasadena, Garmisch, Sodankyla, Anemyondo, Hefei, and Burgos) and SH (Darwin, Wollongong, Reunion, Ascension, and Manaus) are provided in Figure S2. The hemispheric differences of CO, $CO_2$, and $CH_4$ are evident among TCCON locations, like that is observed in Figure 1 from satellite retrievals. The corresponding mean magnitude and the corresponding variability of the abundance during 2012-2019 period is also provided in Table 2. The mean abundance of CO ranges from 118.3±13.5 ppb in Hefei to 59.7±7.8 ppb in Wollongong. The observed abundance of CO is higher at measurement sites in the Southeast Asian regions (Hefei, Anmyeondo, and Burgos) in comparison to other selected sites. These values are consistent with literature that reported higher emissions over Southeast Asian regions (Tang et al., 2019; Zhang et al., 2020), especially from fossil fuel, coal, agriculture activities and wetlands (Tang et al., 2019).

We also see a decreasing trend in CO in most of the selected TCCON sites (-0.20 to -0.98 % ppb/year), except at Ascension (3.51±0.43 % ppb/year), Pasadena (0.01±0.22 % ppb/year), and Wollongong (0.27±0.35 % ppb/year). This agrees with the long-term decline in the column abundances of global CO reports in the literature (Zhang et al., 2020; Buchholz et al., 2021). The mean abundance of column $CO_2$ varies from 406.8±1.9 ppm in Burgos to 397.4±5.1 ppm in Wollongong. The mean column abundance of $CH_4$ ranges from 1.88± 0.02 ppm in Hefei to 1.77±0.02 ppm in Wollongong. The trend calculated during the 2012-2019 period for the mean column abundance relative to 2012 is provided in Table 2. $CO_2$ and $CH_4$ are showing an increasing trend in all locations. The trend in $CO_2$ is higher over Anmyeondo (0.81± 0.10 % ppm/year), and lower over Ascension, (0.60±0.01 % ppm/year). Similarly, Sodankyla (0.48±0.02 % ppm/year) shows a higher trend in $CH_4$ and a lower trend in Anmyeondo (0.21± 0.15 % ppm/year). This is may due to differences in the distribution of sources and/or sinks across these sites as described in section 2.1. Retrievals from TROPOMI CO, for example, show a southward transport of enhanced CO concentrations over Atlantic Ocean originating from fires in North Africa (Borsdorff et al., 2018). The high CO polluted air (~116 ppb) captured over Ascension Island in TROPOMI agrees with TCCON site in Ascension (Feist et al., 2014 and Borsdorff et al., 2018). The higher trend in CO and a lower trend in $CO_2$ over Ascension may be attributed to a decrease in sources (reduced respiration, increase in lower quality fossil-fuels) or an increase in sinks (enhanced photosynthesis) over the African region. For example, Hickman et al. (2021) reported an increasing trend in CO and reduction in $NO_2$ burden over north equatorial Africa and attributed this to a decline in biomass burning due to emissions from a woodier biome. We note however that detailed source and sink analysis (including transport patterns over Ascension) is needed to better understand this high CO and low $CO_2$ trends in this region.




**Table 2: Mean and standard deviation, trend, amplitude, and co-variation of CO, $CO_2$, and $CH_4$ over Pasadena, Ascension, Manaus, Garmisch, Sodankyla, Anmyeondo, Burgos, Hefei, Darwin, Wollongong, and Reunion. The correlations between the species are shown using a linear (Pearson's correlation) and a non-linear (mutual information/MI) metric.**

| Locations | Pasadena | Ascension | Manaus | Garmisch | Sodankyla | Anmyeondo | Burgos | Hefei | Darwin | Wollongong | Reunion |
|---|---|---|---|---|---|---|---|---|---|---|---|
| CO | 93.5 ±11.5 | 84.4 ±10.3 | 94.0 ±12.0 | 86.4 ±9.1 | 88.5 ±11.1 | 104.8 ±10.8 | 84.5 ±11.9 | 118.3 ±13.5 | 72.0 ±12.2 | 59.7 ±7.8 | 67.3 ±9.1 |
| $CO_2$ | 403.5 ±5.5 | 398.3 ±4.0 | 398.6 ±1.2 | 400.5 ±6.0 | 399.6 ±6.7 | 403.3 ±3.8 | 406.8 ±1.9 | 404.5 ±2.8 | 397.8 ±5.1 | 397.4 ±5.1 | 397.8 ±4.5 |
| $CH_4$ | 1.83 ±0.02 | 1.81 ±0.01 | 1.83 ±0.01 | 1.82 ±0.02 | 1.81 ±0.02 | 1.85 ±0.01 | 1.85 ±0.02 | 1.88 ± 0.02 | 1.79± 0.02 | 1.77 ±0.02 | 1.78 ±0.01 |
| Trend in CO | 0.01 ±0.22 | 3.51 ±0.43 | | -0.00 ±0.14 | -0.53 ± 0.22 | -0.31 ±1.64 | | | -0.98 ±0.64 | 0.27 ±0.35 | -0.20 ±0.41 |
| Trend in $CO_2$ | 0.68 ±0.01 | 0.60 ±0.01 | | 0.66 ±0.01 | 0.69 ±0.02 | 0.81 ± 0.10 | | | 0.66 ±0.01 | 0.64 ±0.01 | 0.63 ±0.01 |
| Trend in $CH_4$ | 0.36 ±0.03 | 0.45 ±0.01 | | 0.47 ±0.02 | 0.48 ±0.02 | 0.21 ±0.15 | | | 0.45 ±0.02 | 0.44 ±0.01 | 0.45 ±0.01 |
| Seasonal amplitude CO | 36.0 ±4.5 | 35.3 ±3.1 | | 33.2 ±8.5 | 37.2 ±3.9 | 16.4 ±0.0 | 27.4 ±15.5 | 38.3 ±0.0 | 33.7 ±10.2 | 33.2 ±8.5 | 33.7 ±9.2 |
| Seasonal amplitude $CO_2$ | 4.7 ±1.3 | 3.6 ±1.9 | | 4.6 ±1.2 | 4.6± 1.4 | 3.6 ±0.0 | 4.6 ±1.4 | 5.6 ±0.0 | 4.4 ±1.4 | 4.6 ±1.2 | 4.4 ±1.3 |
| Seasonal amplitude $CH_4$ | 0.03 ±0.01 | 0.03 ±0.01 | | 0.03 ±0.01 | 0.03 ±0.00 | 0.01 ±0.00 | 0.02 ±0.01 | 0.03 ±0.00 | 0.03 ±0.01 | 0.03 ±0.01 | 0.03 ±0.01 |
| Correlation of CO:$CO_2$ | 0.11 | 0.44 | -0.66 | 0.12 | 0.19 | 0.29 | 0.42 | 0.41 | 0.03 | 0.13 | 0.20 |
| MI of CO:$CO_2$ | 0.11 | 0.18 | 0.39 | 0.27 | 0.42 | 0.48 | 0.30 | 0.35 | 0.19 | 0.16 | 0.17 |
| Correlation of $CH_4$:$CO_2$ | 0.62 | 0.88 | 0.36 | 0.75 | 0.52 | -0.04 | -0.03 | -0.13 | 0.93 | 0.80 | 0.88 |
| MI of $CH_4$:$CO_2$ | 0.30 | 0.68 | 0.19 | 0.55 | 0.46 | 0.55 | 0.22 | 0.37 | 1.00 | 0.54 | 0.73 |
| Correlation of CO:$CH_4$ | 0.20 | 0.48 | -0.05 | -0.06 | -0.19 | 0.18 | 0.38 | 0.23 | 0.1 | 0.39 | 0.26 |
| MI of CO:$CH_4$ | 0.11 | 0.17 | 0.39 | 0.16 | 0.24 | 0.36 | 0.42 | 0.31 | 0.12 | 0.17 | 0.10 |

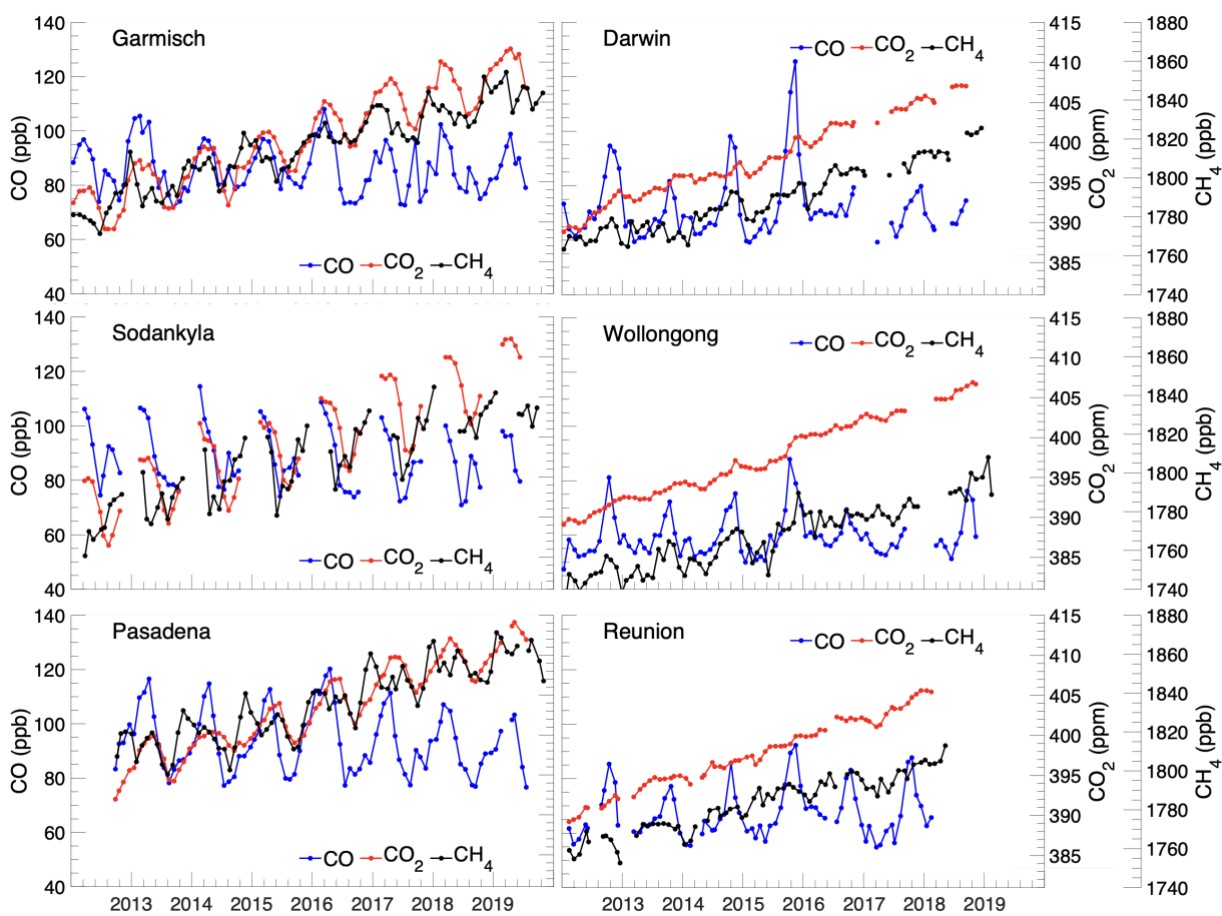

**Figure 2: Monthly variation of TCCON CO, CO₂, and CH₄ over Garmisch, Darwin, Sodankyla, Wollongong, Pasadena, and Reunion during 2012 to 2019.**

To elucidate the dependence of similar variations and/or similar sources of origin, we also show in Figure 3 the joint probability density distribution (pdf) between CO and $CO_2$, CO and $CH_4$, as well as $CO_2$ and $CH_4$. We also provide estimates of the associated dependencies (linear vs non-linear) among these species for the whole analysis period as presented in Table 2. The linear relationship is quantified using the Pearson's correlation while the non-linear dependency is estimated using mutual information (Kraskov et al., 2004). Consistent correlations across all three species suggests a similar source of origin, seen in the strong linear correlation across the species in Ascension and strong non-linear correlation across the species in Anmyeondo and Hefei. Strong dependencies are observed among $CO_2$ and $CH_4$ in most locations, where the correlations are higher than the ones between CO and $CO_2$ and CO and $CH_4$. This is also seen in the joint distributions shown in Figure 3 where the relationship between $CO_2$ and $CH_4$ is more apparent compared to others and point towards a shared signature from biospheric/natural and anthropogenic activities leading to a strong relationship between $CO_2$ and $CH_4$. The differences





observed between the non-linear and linear dependencies highlight the complexity of the relationship between the species and can be associated with the presence of daily variation in the sources and sinks, seasonality, differences in the lifetime of the species, as well as changes in the background present in the entire analysis period. We further investigate the variations in corresponding enhancement ratios in the next section to understand these differences.

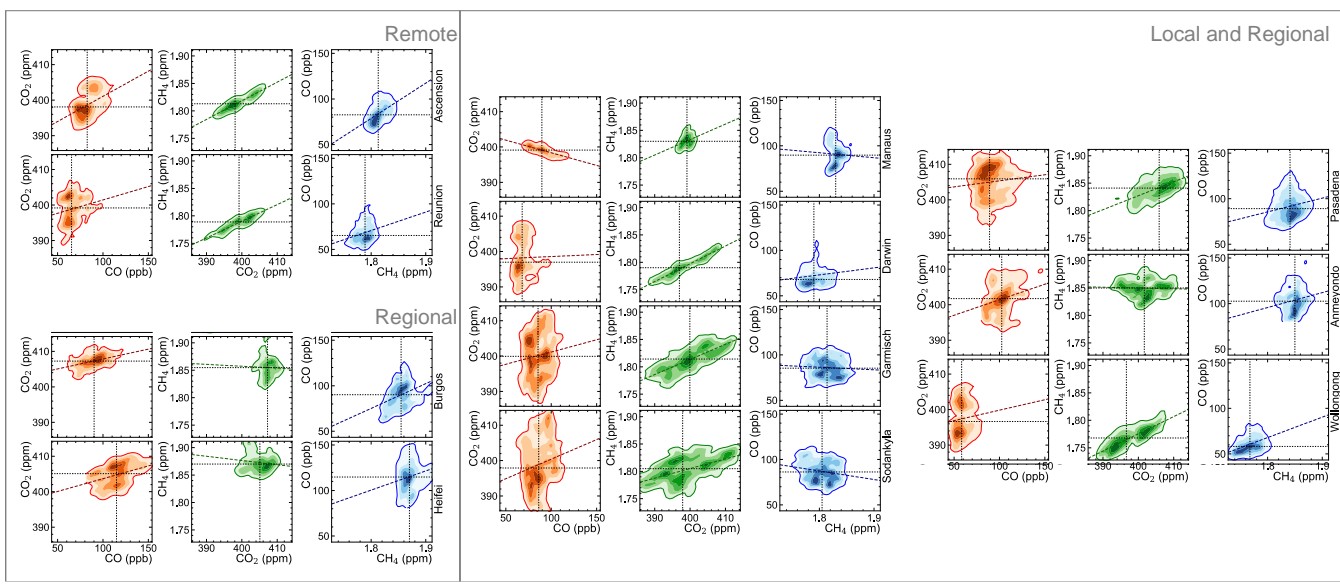

**Figure 3: Joint probability distributions between CO and CO₂ (orange), CO₂ and CH₄ (green) and CH₄ and CO (blue) using daily values across 11 TCCON sites chosen for this study. The sites are grouped according to the site type and source influence on the species in these regions. CO is shown in ppb, whereas CO₂ and CH₄ have units in ppm. The straight lines denote the best-fit line from linear regression.**

**3.2. Enhancement ratio of CO, CO₂, and CH₄: Regional and local contributions and associated seasonality**

**Enhancement Ratios.** Figure 4 shows the mean variation of these enhancement ratios in CO/CO₂, CH₄/CO₂ and CO/CH₄. Note that these ratios are calculated monthly across the daily data based on the methods explained in Section 2.2. The bulk enhancement ratio (BERr), which accounts for the total emission sources, sinks, and other contributions to observed abundances, is higher for all species in all measurement sites in comparison to the local enhancement ratios (AERa and AERr).
Regionally, BERr in CH₄/CO₂ is maximum over the Southeast Asian region (Anmyeondo, Burgos and Hefei) followed by the sites in SH locations (Darwin, Wollongong, Reunion, Ascension) when compared to other NH sites. This higher value of BERr in Southeast Asian region follows the regional maximum of CO and CO₂ mixing ratios described in section 3.1 and shown in Figure S2. Similar is the case for the regional site variation of BERr in CO/CH₄. The value of CH₄/CO₂ from BERr is highest over Burgos and Wollongong followed by Garmisch, Sodankyla, Anmyeondo, and Pasadena. Relative differences can be
observed between the correlations across the species and BERr suggesting more complex mixtures of the sources and sinks of these species at these sites.



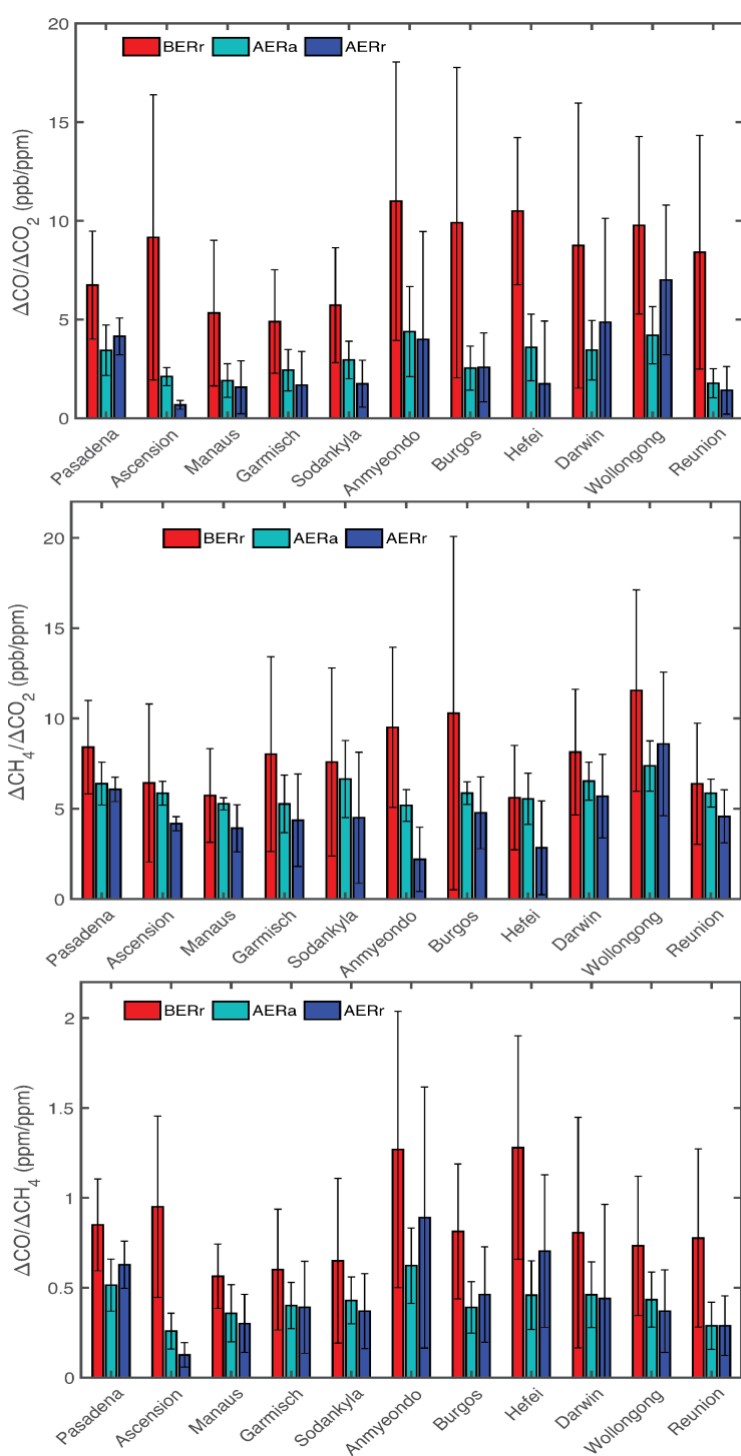

**Figure 4: Mean variation of enhancement ratios calculated as Bulk Enhancement Ratio (BERr), Anomaly Enhancement Ratio (AERa), and Anomaly Enhancement Regression Ratio (AERr) of CO/CO₂, CH₄/CO₂ and CO/CH₄ during 2012-2019 over Pasadena, Ascension, Manaus, Garmisch, Sodankyla, Anmyeondo, Burgos, Hefei, Darwin, Wollongong, and Reunion.**





We note that the enhancement ratios derived in this work is within the range of ratio estimates reported in literature (Wunch et al., 2009; Wennberg et al., 2012; Silva et al., 2013; Buchholz et al., 2016; Hedelius et al., 2018; Bukosa et al., 2019). In Pasadena, Silva et al. (2013) reported an enhancement ratio in $CO/CO_2$ of about 9.3 -13.5 ppb/ppm based on MOPITTv5 and ACOS2.9/GOSAT $CO_2$ data, while Wunch et al. (2009) and Wennberg et al. (2012) reported 11 ppb/ppm and 8.4 ppb/ppm,

335 respectively, along with the more recent study by Hedelius et al. (2018) which reported 7.1 to 7.5 ppb/ppm. Buchholz et al. (2016) and Bukosa et al. (2019) reported a range of ratios of about 1.3-37.4 ppb/ppm in $CO/CO_2$, 9.8-61 ppb/ppm in $CH_4/CO_2$ and 0.3-13 ppb/ppb in $CH_4/CO$ over Australia. While generally consistent, our estimates also show that the range of ratios reported in these studies can vary (as can be expected) depending on the dominant processes (natural and/or anthropogenic) driving species abundance.

340 **Regional and Local Contributions.** Additionally, the differences in the enhancement ratio from BERr, AERa, and AERr in Figure 4 can be indicative of different regional and local influences. As described in Section 2.2, the enhancement ratio calculated from the regression slope of the anomalies (AERr) represents a local enhancement ratio, where the associated regional enhancement ratio can then be derived by subtracting AERr from BERr (i.e., regional=bulk – local). Figure 5 shows the average seasonal variation of the regional (BERr - AERr) and local enhancement ratios (AERr) for each species. This

345 reveals how the contribution and influence of regional and local enhancement ratios in the bulk ratio vary seasonally. The seasonal variations calculated for DJF should read as Winter in NH and Summer in SH, MAM months as Spring in NH and Fall in SH, JJA months as Summer in NH and Winter in SH and SON months as Fall in NH and Spring in SH. The corresponding number of months available to generate the average seasonal variation of regional and local enhancement ratio is provided in supplementary material (Table S1 and S2). Note that for sites like Sodankyla, there are only 4 data points for

350 seasonal averaging during winter months due to limited measurements in this period.

We see in Figure 5 that the seasonal variation of regional and local enhancement ratios at different measurement sites reveals the presence of seasonally varying driving factors in the bulk enhancement ratios. The local enhancement ratio appears to dominate over the regional ratios for Pasadena in all seasons and relative to the regional ratio, the magnitude of local enhancement ratios in $CO/CO_2$ and $CO/CH_4$ are more significant during Fall. The lower regional enhancement ratio during

355 Fall may be due to the poor dependency between transported $CH_4$ or $CO_2$ coming from biospheric sources or any non-combustion sources of CO. This is evident in Figure S2 which shows a significant peak in the abundance of $CO_2$ during Fall months over Pasadena, but not in CO. Furthermore, the low value of regional enhancement ratio in $CH_4/CO_2$ during Summer over Pasadena may be associated with the poor correlation from independent sources or from biospheric sinks of $CO_2$ (see Tables S1 and S2). Similar seasonal variation is observed at Wollongong where it shows a dominant influence of local

360 enhancements of species ratios for most of the seasons. Relative to the regional ratio, the magnitude of local enhancement ratio in $CH_4/CO_2$ is more significant during the months of DJF, which is the summer season in SH. The seasonal variation of $CO/CH_4$ follows a different pattern in Wollongong with the regional influence dominating for all seasons except JJA (winter in SH).




The seasonal variation of species enhancement ratio in $CH_4/CO_2$ and $CO/CH_4$ at Darwin follows similar variations as that in Wollongong although there are differences in absolute magnitude. The regional enhancement ratio in $CO/CO_2$ dominates during DJF (summer) and SON (spring) months at Darwin whereas the local enhancement ratio dominates in other seasons. A large difference of about 10 ppb/ppm is also observed between local and regional enhancement ratio in $CO/CO_2$ during JJA (winter) months.

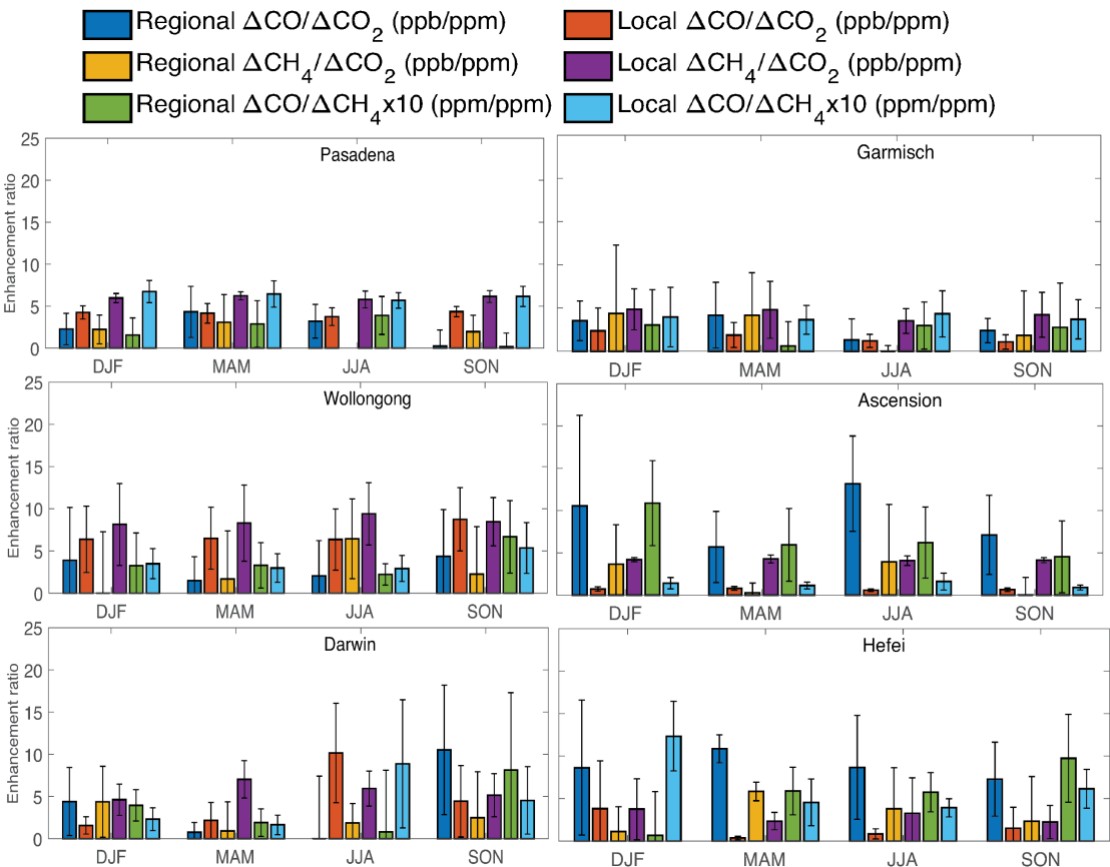

**Figure 5: Average seasonal variation of regional and local enhancement ratio in $CO/CO_2$, $CH_4/CO_2$ and $CO/CH_4$ during 2012-2019 over Pasadena, Garmisch, Wollongong, Ascension, Darwin, and Hefei.**

Furthermore, in Ascension, the influence of regional enhancement ratios in $CO/CO_2$ and $CO/CH_4$ is high during all seasons whereas the seasonal variation in $CH_4/CO_2$ shows a different pattern. Except in Spring (SON) and Fall (MAM), the seasonal influence of the regional and local enhancement ratio in $CH_4/CO_2$ is comparable. The low values of regional enhancement in $CH_4/CO_2$ during Spring and Fall may be associated with the poor correlation from independent sources or from biospheric sources of $CO_2$. The seasonal variation of enhancement ratio at Manaus and Reunion follows this characteristic as well (shown in Figure S3). The relative importance of regional and local enhancement ratio varies among species in Garmisch and Sodankyla. The regional enhancement ratio in $CO/CO_2$ and local enhancement ratio in $CO/CH_4$ ratio dominate for all seasons



at Garmisch (Figure 5) and Sodankyla (Figure S3) while the local enhancement ratio in $CH_4/CO_2$ dominates during JJA (winter) and SON (spring) months compared to other seasons over these sites. Finally, irrespective of the season, regional

enhancements in $CO/CO_2$ dominate at Hefei and Burgos (Figure S3) while the same is true in $CH_4/CO_2$ at Anmyeondo (Figure S3). The local enhancement ratio in $CH_4/CO_2$ and $CO/CH_4$ dominates only during DJF (winter) at Hefei, while local enhancement ratio in $CO/CH_4$ dominates for all seasons at Anmyeondo except fall (SON). The local enhancement ratio in $CO/CH_4$ also dominates regardless of season at Burgos.

The average relative contribution of local and regional enhancement ratio towards the bulk enhancement ratio at the

385 measurement site is provided in Table 3. The relative contribution of the regional and local enhancement ratio is calculated as $\frac{BERr-AERr}{BERr}$ and $\frac{AERr}{BERr}$ respectively. A clear difference is observed in the contribution of the local and regional enhancement ratios across each measurement site and among species. Locations like Pasadena and Wollongong show the dominant local influence for $CO/CO_2$ whereas the rest of the locations report significant regional influences. This regional contribution in $CO/CO_2$ to the bulk enhancement ratio is highest over Ascension followed by Burgos (>80%). This can be attributed to the fact that

Ascension is a remote location and the sharp rise in the column abundance of CO at Ascension can be associated to a rise in transported CO from the nearby African region. Previous studies over Burgos and vicinity also reported enhanced CO and $CH_4$ due to transport of emissions from East Asia (Velazco et al., 2017; Hilario et al., 2021). This inference is in support of the location features provided in Section 2.1 and source information as reported in previous studies. The contribution of regional enhancement ratios dominates over Manaus, Anmyeondo, Sodankyla, Hefei and Burgos to the bulk enhancement ratio in

$CH_4/CO_2$ while the remaining sites report dominance of its local enhancement ratio. Except for Ascension, Manaus, Darwin, Anmyeondo, and Reunion, the contribution of local enhancement ratio in $CO/CH_4$ is higher than the regional at all other measurement sites.

With the difference in the contributions of regional and local enhancement ratios, we can also derive the enhancement of each species due to these regional and local enhancements. The mean enhancement $(\overline{\Delta C_Y})_i$ of a species, $Y$ for example, can be

calculated as the product of $\left(\frac{\Delta C_Y}{\Delta C_X}\right)_i$ and $C'_X$, where $i$ is either $R$=BERr-AERr or $L$ =AERr, representing the regional ($R$) and local ($L$) enhancement ratio respectively and $C'_X$ is the anomaly of species $X$ calculated using Method 2 (AERa). That is, the regional ($R$) enhancement of CO for this example can be derived from the enhancement ratio in $CO/CO_2$ as: $\Delta C_{Y|X}{}^R = \left[\left(\frac{\Delta C_Y}{\Delta C_X}\right)_R \cdot C'_X\right]$ and similarly from the enhancement ratio in $CO/CH_4$ as: $\Delta C_{Y|Z}{}^R = \left[\left(\frac{\Delta C_Y}{\Delta C_Z}\right)_R \cdot C'_Z\right]$. We then take the mean of two enhancements ($\Delta C_{Y|X}{}^R$ and $\Delta C_{Y|Z}{}^R$) for species $Y$ to account for species variations. Similar calculations are carried out for

local ($L$) enhancements. The average variation of the enhancements of CO, $CO_2$, and $CH_4$ from the local and regional enhancement is provided in Figure S4. A large difference (10-28 ppb) is observed in the relative increase of CO between regional and local enhancements over Burgos, Ascension, and Reunion. The relative increase of $CO_2$ at Sodankyla, Anmyeondo, and Burgos show dominance of local enhancements while the remaining locations show higher importance of





regional processes. Except at Ascension and Anmyeondo, all other measurement sites show that the relative rise in $CH_4$ is
410 coming from regional processes. The difference in relative increase in $CO_2$ and $CH_4$ between regional and local enhancements
is less in most of the locations compared to the corresponding relative increase in CO. This smaller difference in the relative
increase can be attributed to the long lifetime, uniform mixing characteristic and the large background value of $CO_2$ and $CH_4$
compared to that of CO in the atmosphere. The different process or source types leading to this regional variation and
seasonality in the local and regional enhancement ratio is further analysed using the scatterplots of multiple species ratios in
the next section.

**Table 3: Percent contribution of regional and local enhancements to the ratio of CO/CO₂, CH₄/CO₂ and CO/CH₄ during 2012-2019 over Pasadena, Ascension, Manaus, Garmisch, Sodankyla, Anmyeondo, Burgos, Hefei, Darwin, Wollongong, and Reunion.**

| | CO/CO₂ | | CH₄/CO₂ | | CO/CH₄ | |
|---|---|---|---|---|---|---|
| Location | Local (%) | Regional (%) | Local (%) | Regional (%) | Local (%) | Regional (%) |
| Pasadena | 63.09 | 36.91 | 72.23 | 27.77 | 71.04 | 28.96 |
| Ascension | 11.99 | 88.01 | 59.96 | 40.04 | 16.65 | 83.35 |
| Manaus | 32.34 | 67.66 | 44.57 | 55.43 | 48.51 | 51.49 |
| Garmisch | 33.46 | 66.54 | 50.64 | 49.36 | 51.78 | 48.22 |
| Sodankyla | 30.43 | 69.57 | 41.65 | 58.35 | 52.84 | 47.16 |
| Anmyeondo | 29.35 | 70.65 | 19.92 | 80.08 | 40.84 | 59.16 |
| Burgos | 14.37 | 85.63 | 41.17 | 58.83 | 51.53 | 48.47 |
| Hefei | 27.28 | 72.72 | 49.97 | 50.03 | 51.22 | 48.78 |
| Darwin | 41.05 | 58.95 | 59.14 | 40.86 | 41.83 | 58.17 |
| Wollongong | 59.64 | 40.36 | 58.94 | 41.06 | 46.11 | 53.89 |
| Reunion | 24.56 | 75.44 | 58.35 | 41.65 | 41.93 | 58.07 |

**3.3. Inferring dominant process contribution from multi-species enhancement ratios**

Figure 6 shows the scatter plot of the ratio in $CO/CO_2$ vs $CO/CH_4$ and $CH_4/CO_2$ vs $CH_4/CO$ for regional and local
enhancements. We use the relationship of the multi-species ratios ($CO/CO_2$ vs $CO/CH_4$ and $CH_4/CO_2$ vs $CH_4/CO$) to
qualitatively infer the processes influencing the regional and local enhancements ratios at each measurement site. For example,
high temperature/more-efficient combustion processes lead to the emission of more $CO_2$ compared to CO and low-temperature
combustion produces more CO (Silva and Arellano, 2017). Similarly, activities associated with the extraction of coal, use and
distribution of natural gas, wetland, rice cultivation, landfill, and livestock result in higher emission of $CH_4$ compared to
emissions of CO and $CO_2$. Lower (higher) ratio values of both $CO/CO_2$ vs $CO/CH_4$ in the scatter plots can be related to





processes emitting lower (higher) CO. Similar approach is applied for ratio variations in CH$_4$/CO$_2$ vs CH$_4$/CO. A summary of these categories for both regional and local enhancements are listed in Table 4 and 5.

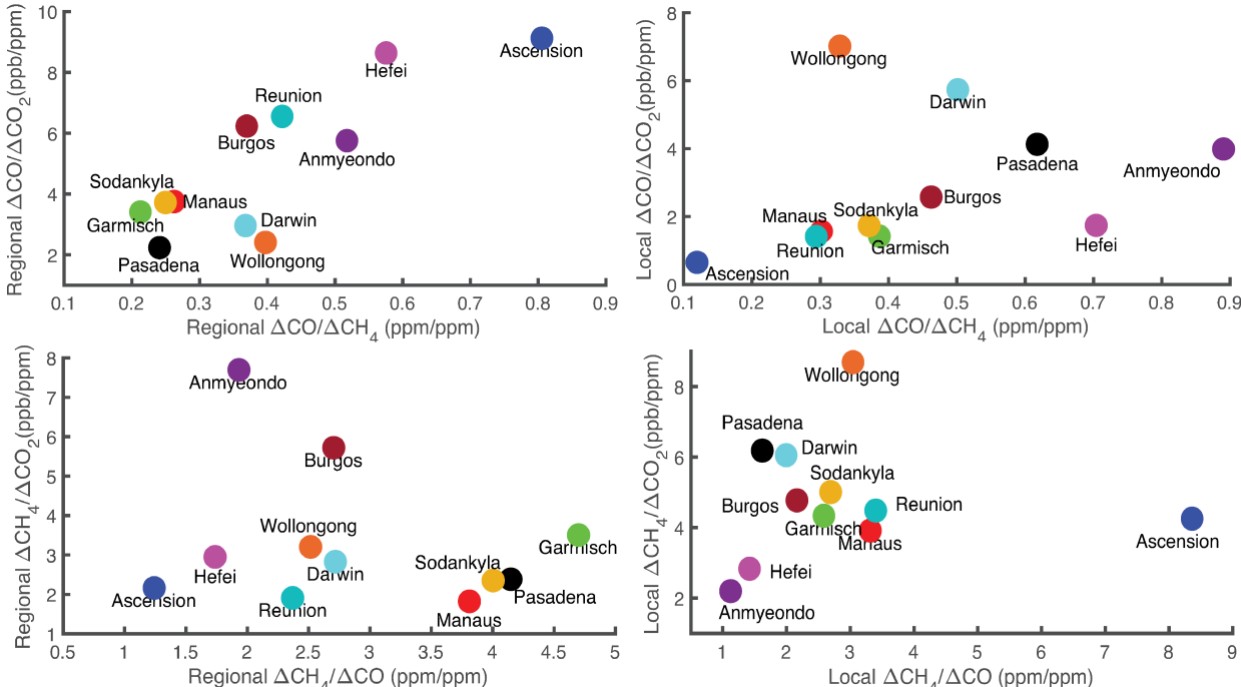

**Figure 6: Scatter plot of average regional (left column) and local (right column) enhancement ratios in CO/CO$_2$ vs CO/CH$_4$ (top row) and CH$_4$/CO$_2$ and CH$_4$/CO (bottom row) during 2012-2019.**

The scatter plot of regional enhancement ratio of species at Pasadena, Manaus, Garmisch, Sodankyla, Darwin, and Wollongong show relatively low value of CO/CO$_2$ vs CO/CH$_4$ and medium/high value of CH$_4$/CO$_2$ vs CH$_4$/CO. The regional enhancement

ratio showed a value between 2.24 and 3.75 ppb/ppm for CO/CO$_2$, between 1.83 to 3.51 ppb/ppm for CH$_4$/CO$_2$ and 3.81 to 4.69 ppm/ppm for CH$_4$/CO over these regions. This pattern can suggest a dominant process (or a combination of) that is characterized by low CO and high CH$_4$ and/or CO$_2$ emissions from natural and biospheric sources, and/or anthropogenic sources with high activity and efficiency. These values fall within the range of previously reported ratios for a mixture of natural and anthropogenic emissions (2-6 ppb/ppb for CH$_4$/CO$_2$ and 3.3-8 ppb/ppm for CO/CO$_2$, Bukosa et al., 2019). The

location features of these measurement sites provided in Section 2.1 also support this result.





**Table 4: Regional process inference based on the ratio of CO/CO₂ vs CO/CH₄ and CH₄/CO₂ and CH₄/CO over Pasadena, Ascension, Manaus, Garmisch, Sodankyla, Anmyeondo, Burgos, Hefei, Darwin, Wollongong, and Reunion.**

| Location | CO/CO$_2$ vs CO/CH$_4$ | CH$_4$/CO$_2$ vs CH$_4$/CO | Regional Process/Source Type |
|---|---|---|---|
| Pasadena | 2.24 vs 0.24 | 2.39 vs 4.15 | Biogenic/biospheric and some combustion |
| Ascension | 9.12 vs 0.805 | 2.16 vs 1.24 | Combustion processes (fires) |
| Manaus | 3.75 vs 0.262 | 1.83 vs 3.81 | Biogenic/Biospheric and some combustion |
| Garmisch | 3.41 vs 0.213 | 3.51 vs 4.69 | Biospheric/Wetland [or other CH$_4$ sources] |
| Sodankyla | 3.72 vs 0.249 | 2.35 vs 4.00 | Biospheric/ Wetland [or other CH$_4$ sources] |
| Anmyeondo | 5.76 vs 0.518 | 7.69 vs 1.93 | High temp combustion/Bio-fuel combustion |
| Burgos | 6.23 vs 0.369 | 5.72 vs 2.71 | Biofuel, coal/some combustion |
| Hefei | 8.64 vs 0.575 | 2.95 vs 1.74 | Low temp combustion (biomass burning) |
| Darwin | 2.96 vs 0.368 | 2.83 vs 2.72 | Biospheric or fires (mixed) |
| Wollongong | 2.41 vs 0.397 | 3.21 vs 2.52 | Biogenic, Bio-fuel combustion (or mixed) |
| Reunion | 6.55 vs 0.422 | 1.91 vs 2.37 | Biospheric/Combustion |

**Table 5: Local process inference based on the ratio of CO/CO₂ vs CO/CH₄ and CH₄/CO₂ and CH₄/CO over Pasadena, Ascension, Manaus, Garmisch, Sodankyla, Anmyeondo, Burgos, Hefei, Darwin, Wollongong, and Reunion.**

| Location | CO/CO$_2$ vs CO/CH$_4$ | CH$_4$/CO$_2$ vs CH$_4$/CO | Local Process/Source Type |
|---|---|---|---|
| Pasadena | 4.13 vs 0.617 | 6.18 vs 1.62 | Biogenic/ Bio-fuel combustion (or fires) |
| Ascension | 0.653 0.119 | 4.26 vs 8.66 | Non-combustion |
| Manaus | 1.57 vs 0.302 | 3.92 vs 3.31 | Biogenic/Biospheric or other combustion |
| Garmisch | 1.42 vs 0.387 | 4.34 vs 2.59 | Biospheric/Biogenic fires |
| Sodankyla | 1.74 vs 0.372 | 5.00 vs 2.69 | Biospheric/Remote |
| Anmyeondo | 3.99 vs 0.890 | 2.20 vs 1.12 | Low temp combustion/ Biofuel combustion |
| Burgos | 2.58 vs 0.462 | 4.78 vs 2.16 | Biospheric and some combustion |
| Hefei | 1.74 vs 0.704 | 2.84 vs 1.42 | Low temp combustion/Biofuel |
| Darwin | 5.73 vs 0.501 | 6.06 vs 1.99 | Biospheric and some combustion |
| Wollongong | 7.02 vs 0.329 | 8.69 vs 3.04 | Biogenic, Bio-fuel combustion (or fires) |
| Reunion | 1.41 vs 0.294 | 4.49 vs 3.39 | Biospheric/ Biogenic fires |

A relatively high/medium value of CO/CO₂ (6.55-9.12 ppb/ppm) vs CO/CH₄ ratio and relatively low value of CH₄/CO₂ (1.91-2.95 ppb/ppm) vs CH₄/CO (1.24-2.37 ppb/ppm) ratio can be seen in Reunion, Ascension, and Hefei. This variation appears to



suggest the presence of low-temperature combustion processes (i.e., biomass burning especially smouldering fires) emitting more CO. A study by Bremer et al. (2004) attributed the enhancement in MOPITT-based CO column abundance at Ascension to Sub-Saharan biomass burning emissions while Zhou et al. (2018) reported that the seasonality of CO at two sites, St Denis and Maido (in Reunion), is primarily driven by biomass burning emissions in Africa and South America. Wang et al. (2017) also reported an enhancement ratio of 5.6 ppb/ppm for $CO/CO_2$ at Hefei during October 2014 and recognized incomplete

combustion of fossil fuels as the main source of CO in this area. The relatively medium value of $CO/CO_2$ (5.76 and 6.23 ppb/ppm) vs $CO/CH_4$ and high $CH_4/CO_2$ (7.69 and 5.72 ppb/ppm) vs $CH_4/CO$ (1.93 and 2.71 ppm/ppm) suggest the presence of fossil fuel emissions, coal/biofuel processes, agriculture, or wetland emissions over Anmyeondo and Burgos. The ratio is close to the range of ratios of 3.3-8 ppb/ppm for $CO/CO_2$ and 1.6-4.2 ppb/ppb for $CH_4/CO$ reported in emissions of mixed anthropogenic sources from rural and urban areas (Bukosa et al., 2019). Initial analysis of TCCON data in Burgos by Velazco

et al. (2017) suggested that the enhancement in CO over the northern part of the Philippines is mostly from fossil fuel emissions, which is dominated by transported emissions from East Asia, and have little influence from biomass burning, which can be large over the southern part of the region (Edwards et al., 2021).

The scatter plot of local enhancement ratio over Wollongong conveys a relatively high/medium ratio in $CO/CO_2$ (7.02 ppb/ppm) vs $CO/CH_4$ and relatively high/medium ratio in $CH_4/CO_2$ (8.69 ppb/ppm) vs $CH_4/CO$ (3.04 ppm/ppm). This appears

to suggest active low-temperature combustion (biomass burning or fires) producing CO and biofuel combustion or coal activities leading to the production of more $CH_4$. This value is within the range of values reported for mixed anthropogenic emissions in Wollongong (Buchholz et al., 2016). Our estimated value is less than the ratio of 13-61 ppb/ppm in $CH_4/CO_2$ reported in Wollongong for coal mining. This may be due to the impact of mixing (dilution) of other sources. The ratio of 4.13 and 5.73 ppb/ppm in $CO/CO_2$, 6.18 and 6.06 ppb/ppm in $CH_4/CO_2$ and a lower ratio in $CH_4/CO$ (1.62 and 1.99 ppm/ppm)

appears to suggest the presence of mixed emissions from anthropogenic or combustion activities in Pasadena and Darwin. This coincides with reports by Hedelius et al. (2018) of a canyon gas leak and wildfire activities based on a ratio of 7.3 ppb/ppm in $CH_4/CO_2$ and 7.1 ppb/ppm in $CO/CO_2$ in Pasadena. The local enhancement ratio at remaining locations shows a relatively low ratio in $CO/CO_2$ vs $CO/CH_4$ and relatively medium/high ratio in $CH_4/CO_2$ vs $CH_4/CO$, which can indicate dominance of biogenic or non-combustion processes influencing these ratios at these locations. The scatter plots of these enhancement ratios

between species across seasons (Figures S5 to S8) reveal similar results shown in Figure 6, but slight seasonal variations are observed at Hefei, Reunion, Darwin, and Wollongong.

**Comparison with Emission Estimates.** We show in Figure 7 the average contribution (in %) to the emissions of CO, $CO_2$, and $CH_4$ over these measurement sites from the anthropogenic sector as reported in the Copernicus Atmosphere Monitoring Service emission inventory (CAMS v4.1, Granier et al., 2019), and biomass burning sector as reported in the Global Fire

Emission Database (GFED4, Giglio et al., 2013). These emission inventories are utilized for qualitative comparison of local emission sources or processes inferred from the scatterplot relationships of multi-species enhancement ratios (see Table 4 and 5). It has to be noted that most of the emissions from the anthropogenic sector of CAMS have emissions with less temporal





variability compared to seasonal variability, including inter-annual variability of biomass burning emissions from GFED. The average total emissions around the grid location of the TCCON measurement site is also provided in Figure 7.

**Figure 7: Sectoral emission distribution (%) of CO, CO₂, and CH₄ from CAMS anthropogenic emissions (left) and GFED fire (right) at TCCON measurement sites during 2012 - 2019. Corresponding total emissions are indicated in the secondary (right) y-axis.**

Regionally, the anthropogenic and fire emission sectors dominate over Hefei, Wollongong, and Darwin compared to other sites (Figure 7). The anthropogenic emission sectors for CO, CO₂, and CH₄ are also significant over Hefei, Pasadena, Wollongong, and Anmyeondo. Residential combustion, industries, power generation, and road transport influence local CO at Hefei. Similarly, residential combustion, industries, and road transport influences local CO in Pasadena whereas in Wollongong CO emissions come from only residential combustion and road transport sectors. A large portion of CO₂ emission in Hefei comes from the power generation sector followed by industries and residential combustion. The major CO₂ emission sectors in Pasadena include industry and residential combustion. Wollongong has CO₂ emissions from the following sectors:



industry, residential combustion, and ships. Note that Hefei, Pasadena, Anmyeondo, and Wollongong have significant emissions of $CH_4$ from anthropogenic sectors. Solid waste, and agricultural soils are the significant emission sectors for $CH_4$ at Hefei. The main sectors for $CH_4$ emissions at Anmyeondo include livestock and agricultural soils. Emissions from fugitives, solid waste and water are significant emitters of $CH_4$ at Wollongong. These mixtures of emission sectors at these sites support the dominant processes identified in the previous section using the correlation of enhancement ratios of these species from TCCON (Figure 6, Table 4, and 5).

The emission from biomass burning is one of the main factors influencing the seasonality and inter-annual variability in the abundance of species. The strong monthly variability of CO and $CO_2$ at Darwin, Wollongong, Reunion, and Pasadena can be attributed to the seasonality of biomass burning emissions (Figure 2 and Figure 6). Agricultural waste burning is the main emission sector for CO, $CO_2$, and $CH_4$ at Hefei. The seasonality of CO, $CO_2$, and $CH_4$ at Wollongong is due to emissions from temperate forest fires (Figure 6) while the biomass burning activity at Darwin, Reunion, and Manaus appears to be dominated by savanna fires followed by agricultural waste burning (Figure 6). Sodankyla, Ascension, and Burgos sites are remote locations and surrounding (local) emissions are therefore smaller than that of the other sites. Even though Reunion Island is a relatively small and isolated island, contribution from local biomass burning activity and other anthropogenic sources is found to be considerable.

## 4 Summary and Future Directions

Despite the growing global burden in $CO_2$ and $CH_4$, current measurements of total column $CO_2$ and $CH_4$ provide a limited verifiable capability in identifying and quantifying specific types of their corresponding sources and sinks. In addition to the lack of vertical information from these column measurements, the diffusive nature of the atmosphere (mixing air masses influenced by spatially and temporally heterogenous sources and sinks), make it very challenging to track source type contributions to these observed column abundances. In this work, we combine simultaneous ground-based measurements of total column abundances of $CO_2$ and $CH_4$ with CO to further characterize the associated enhancements in the column abundance of the respective species by taking advantage of their temporal co-variations. A total of 11 sites from Total Carbon Column Observing Network (TCCON), including six stations in NH and five in SH, are selected to investigate associated multi-species patterns during 2012 to 2019 period. We also introduce a combination of established regression and anomaly approaches to derive mean local and bulk enhancement ratios between $CO/CO_2$, $CO/CH_4$ and $CO_2/CH_4$ across each month of daily data. We first derive "bulk" enhancement ratios (BERr) using 3 regression algorithms (ordinary least square, geometric regression, York regression) where we report the BERr as the mean across these algorithms weighted by the associated errors. We also employ a "local" anomaly approach, where observed columns are presubtracted by assumed "background" values. These values are derived as the mean of a) daily anomalies calculated by subtracting the morning from afternoon columns; and b) $5^{th}$ percentile of daily data. The enhancement ratios based on anomalies are derived either from monthly mean ratios (AERa) or regressed slope (AERr) between these anomalies. This combination of approaches allows us to not only account



for the variability on our estimates of enhancement ratios due in fact from the algorithm and assumptions of background values, but also to separate the regional and local influences on these ratios by subtracting BERr ("bulk or global") from AERr or AERr ("local") estimates.

Our results show that: a) estimates of enhancement ratios are within the range of ratio estimates reported in literature; b) regional and local influences to these ratios can be disentangled with resulting values that appear to be physically reasonable relative to current understanding of process drivers at these site locations; and c) multi-species analysis of these enhancement ratios can augment current techniques aimed at characterizing dominant types of sources and sinks influencing observed abundances. We find that Pasadena (Wollongong, Manaus) shows a dominant (moderate) local influence (>60% in Pasadena,

>50% in Wollongong and Manaus) across CO, $CH_4$, and $CO_2$ which appear to come from a mixture of biospheric and combustion activities. In contrast, Anmyeondo show a dominant regional influence (>~60%) across all species, which appear to come from high temperature and/or biofuel combustion activities. Comparable influence of regional and local enhancement is observed in Darwin (biospheric and/or low-temperature combustion) for all species. Interestingly, Sodankyla and Garmisch (mostly biospheric and wetlands), Hefei (low-temperature combustion) and Burgos (biofuel combustion) are characterized by

larger regional influence (~67 for Garmisch, ~70% for Sodankyla, ~73% for Hefei and 86% for Burgos) in $CO/CO_2$ and relatively comparable regional and local influences in $CH_4/CO_2$ and $CO/CH_4$. On the other hand, Ascension shows a large regional influence (>80%) for both $CO/CO_2$ and $CO/CH_4$ indicative of fire activities (high CO). While Ascension is relative characterized as "remote" with little local influence in column CO, it appears to show the impact of long-range transported emissions (most likely fires). Note that column CO can capture this fire signature as opposed to several reports over Ascension

which have indicated that fire plumes from southern Africa cannot be observed from ground-based site in the island. Similar finding is observed in Reunion (albeit not as large regional influence, ~75 in $CO/CO_2$ and ~58% in $CO/CH_4$). As with Ascension, Reunion is on an isolated island and characterized as "remote" but with large presence of combustion (fire) influence as it receives higher amounts of smoke outflows from African fires on its west. These results are qualitatively consistent with corresponding estimates from CAMS and GFED emission inventories.

This work is envisioned to serve as one of the bases for interpreting enhancement ratios derived from current space-borne collocated column measurements of CO, $CO_2$, and/or $CH_4$ (e.g., TROPOMI, GOSAT-2, OCO-2, and OCO-3). The method presented here can also be applied to future geostationary satellites that will provide sub-daily measurements such as GeoCARB (e.g., Moore et al., 2018). Our method provides a preliminary framework towards the evaluation of the enhancement ratios (i.e., species sensitivities) along with the abundances derived from these satellite missions to reduce the discrepancies

between the top-down and bottom-up inversions and emission-based studies, as well as to provide more robust source type attribution of these abundances that otherwise is difficult to obtain by single species analysis alone. The use of enhancement ratios and their separation into regional and local influence allows us to effectively disentangle the source type and transport signatures of these species over the sites, unlike the correlation estimates in Section 3 which do not provide a complete picture considering the diffused (non-linear) behaviour of their sources and sinks. Separating the contributions of megacity emissions



from fire and biogenic sources is a future application of this study. Use of data-driven machine learning regression algorithms can also assist in inferring the contribution from different emission sources.

**Acknowledgements**

This research work is supported by NASA ACMAP Grant (80NSSC19K0947). Dr. Tang is supported by NCAR Advanced Study Program Postdoctoral Fellowship. The TCCON data for total column measurement of CO, $CO_2$ and $CH_4$ at Pasadena,

Ascension, Manaus, Garmisch, Sodankyla, Anmyeondo, Burgos, Hefei, Darwin, Wollongong, and Reunion were obtained from the TCCON Data Archive hosted by CaltechDATA at https://tccondata.org. We also acknowledge the Emission of Atmospheric Compounds and Compilation of the Ancillary Data (ECCAD, https://eccad3.sedoo.fr) for anthropogenic and biomass burning emission data of CO, $CO_2$, and $CH_4$ from the inventories of Copernicus Atmosphere Monitoring Service (CAMS v4.1) and Global Fire Emission Database (GFED4) during 2012-2019 period. This material is partly based upon work

supported by National Center for Atmospheric Research, which is a major facility sponsored by the National Science Foundation under cooperative agreement no. 1852977.

**Data Availability Statement**

The TCCON data were obtained from the TCCON Data Archive hosted by CaltechDATA at https://tccondata.org, while the following supporting datasets were obtained from: Emission of Atmospheric Compounds and Compilation of the Ancillary

Data (ECCAD, https://eccad3.sedoo.fr) for CAMS v4.1 and GFED4; MOPITT from NASA through the Earthdata portal (https://earthdata.nasa.gov), GOSAT-1 from NIES at https://data2.gosat.nies.go.jp, and OCO-2 from NASA through the Goddard Earth Science Data and Information Services Center (https://disc.gsfc.nasa.gov/datasets?keywords=l3co2) for registered users.

**Author Contribution**

Conceptualization: AFAJ, WT; Investigation: KM, VB, CR, and AFAJ; Methodology: KM, VB, AFAJ; Formal Analysis: KM, VB, AFAJ; Data curation: KM, CR; Validation: KM, CR; Visualization: KM, CR; Supervision: AFAJ; Writing- original draft preparation: KM, AFAJ; Writing- review & editing: BG, WT, CR, MAM, JM, YG and AFAJ.

**Competing Interests**

No authors have any competing interests.



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
