# Peer review of "Local and Regional Enhancements of CH4, CO, and CO2 Inferred from TCCON Column Measurements"

_EGUsphere, 2024_

## Author Comment (AC1)

**RC1**: 'Comment on egusphere-2024-705', Anonymous Referee #1, 23 Apr 2024

**General Comments:**

In this study, Mottungan et al. demonstrate the use of correlative measurements of CO, CO2, and CH4 data from eleven TCCON sites to identify local and regional airmass characteristics and estimate the relative contributions of local and regional sources at each TCCON site. They use different regression techniques and enhancement ratio algorithms in the analysis. The paper addresses important science questions, is well-structured, and contains informative figures. It fits into the scope of AMT. The current TCCON GGG2020 data are available for several more years than shown in the analysis, and it might be nice to have the new data included in the analysis. The paper is suitable for publication in AMT after minor revisions, data updation/extension, and technical changes.

We thank the reviewer for the valuable comments that have greatly improved the revised manuscript. We have addressed the major and minor comments (black) below with our response in blue.

We understand that it would be better to expand the analysis using GGG2020 data, we chose not to use the newer version (GGG2020) as it does not include sites like Ascension Island and Anmyeondo across 2012 – 2019 that we used in this study.

**Specific Comments:**

**L91:** This is just a personal preference – maybe equations could be used instead of text to explain the two methods. Some of the text could also be moved to the 'Data and Methods' section.

Thank you for this comment. We have outlined the equations used to calculate the enhancement ratios in Section 2.2 under Data and Methods. We included some text about calculating these ratios in the Introduction to provide a background on how these ratios have been traditionally calculated from previous scientific literature.

The paragraph now reads as follows:

> *The enhancement ratio between species X and Y is calculated by mainly two methods: the first is from a linearly regressed slope of X and Y (Andreae et al., 1988a, 1988b) and the second is by dividing the excess of X by the excess of Y (Andreae and Merlet, 2001) (See Methods 1 and 2 in Sect. 2.2 respectively). The first approach of enhancement ratio estimation using regression slopes is difficult to infer when emitted or locally produced species mix with different air masses (e.g., advection from the nearby sources or mixed air masses) downwind of the dominant source where measurements are made. This is especially the case for vertically integrated quantities like the column measurements (either ground-, aircraft- and satellite-based) (Cheng et al., 2017; Halliday et al., 2019; Tang et al., 2019) where vertical information of the species abundance is practically absent. If the emission or plume concentration is significantly larger than the background, the ratio from the regression slope approach does not change (Brigg et al., 2016). But, when emission of the species mixes with different 'backgrounds' than a relatively uniform field, the abundances of X and Y change due to mixing*

*and/or photochemical loss (Mauzerall et al., 1998; Yokelson et al., 2013; Guyon et al., 2005); thus, making it difficult to track the locally emitted contribution to the observed abundance. The latter approach of using excess of the species requires a proper understanding of the background concentration to derive the excess abundance along with the instantaneous concentration of the species, which is not available in most cases. Vertical and horizontal transport also complicates the interpretation of abundance and assessment of local and regional source influences at a particular location (Chatfield et al., 2020). A combination of these two approaches have also been used in previous studies (Hedelius et al., 2018) (Method 3 in Sect. 2.2). Here, we utilize the column measurements of CO, $CO_2$, and $CH_4$ (denoted as $X_{CO}$, $X_{CO2}$, $X_{CH4}$) from the Total Carbon Column Observing Network (TCCON) (Wunch et al., 2011) to understand these variations in the column abundances.*

**L115:** "While previous studies have used enhancement ratios to examine the source attribution of CH4, CO, and CO2 at regional and/or local scale, we note that few have investigated bulk characteristics on a source type basis using all these 3 species and using these combinations of regression algorithms for globally distributed column-integrated measurements." – Please cite a few examples of studies that have used all 3 species and combinations of regression algorithms.

Thank you for this comment. We have cited a source that previously studied enhancement ratios across the three species CO, $CO_2$, and $CH_4$ in the text as shown below. We wanted to clarify that the study outlined in our manuscript is a first of its kind to 1) use different approaches to calculating enhancement ratios across the three species and different geographical sites, and 2) use a combination of regression algorithms to assist in these calculations. We understand that the original text might have been misleading and have thus corrected it as shown below.

*While previous studies have used enhancement ratios to examine the source attribution of $CH_4$, CO, and $CO_2$ at regional and/or local scale (Bukosa et al., 2019), the novelty of this study lies in investigating the bulk characteristics on a source type basis using all three species and using a combination of regression algorithms for globally distributed column-integrated measurements.*

**L123 and Table 1:** The TCCON data are used from 2012 to 2019, as also mentioned in Table 1. The Public TCCON data (GGG2020) are currently available until 2023 for Pasadena/Caltech, Garmisch, Sodankyla, Burgos, and Wollongong. The data are available until 2022 for Hefei and Darwin and until 2020 for Reunion. In Table 1, the data for these sites are listed for shorter time periods (and in Figure S1). I am curious if the analysis can be expanded and updated to include the additional years of available data.

Thank you for this comment. Since we are using GGG2014 TCCON data, we are limited to the period 2012 to 2019 for the chosen sites. While it would be better to expand the analysis using GGG2020 data, we chose not to use the newer version (GGG2020) as it does not include sites like Ascension Island and Anmyeondo, and using GGG2014 would help maintain the temporal continuity across 2012 – 2019 and across selected sites.

The data link provided in the 'Data Availability Statement' points to the TCCON website, which currently does not list the TCCON data for Ascension Island and Anmyeondo (for GGG2020).

Thank you for this comment. We have updated the link in the Data Availability Statement that points to the GGG2014 dataset (containing Anmyeondo and Ascension).

Please indicate the data version used. Is it GGG2014 (data for Ascension Island and Anmyeondo are available) or GGG2020? In Table 1, the References are used for both data versions (e.g., Sodankyla – Kivi et al. (2014) for GGG2014 and Kivi et al. (2022) for GGG2020).

Thank you for pointing this out. We are using GGG2014 for this dataset and have updated Table 1 to reflect the same. We have also updated the references in Table 1 to reflect the GGG2014 dataset.

Oh et al. (2018) are cited in Table 1. Please add it to the References section.

Thank you for pointing this out. We have updated the references section. We use this reference in line 302 in Section 2.1 instead of Table 1.

Liu et al. (2018) are listed in the References section but not cited in the manuscript.

Thank you for pointing this out. We have revised the text accordingly and cited this reference.

**L124:** If the data version is GGG2020, please consider citing Laughner et al. (2023) (https://doi.org/10.5194/amt-16-1121-2023) and https://doi.org/10.5194/essd-2023-331.

Thank you for the suggestion to cite the reference for GGG2020. We used the GG2014 dataset for this manuscript and have updated the text to reflect the correct references.

**Figure 1:** Please consider listing location names on all sub-plots/panels. The background colors make it difficult to focus on the makers. Please consider changing the maker type and color.

Thank you for pointing this out. We have updated Figure 1 with bigger visible markers for a better comparison between TCCON and satellite data. We have shown the 2012 – 2019 monthly averaged (2015 – 2019 for $X_{CO_2}$) total column measurement for the three species instead of only May 2018 as in the original version.

**L137**: "Qualitatively, MOPITT and GOSAT retrievals show reasonable agreement between the retrieval of CO, CO2, and CH4 column abundance relative to TCCON at these locations." – Please elaborate.

Thank you for pointing this out. We have revised the text to include some quantitative markers for comparison of the column abundances, as shown below,

*$X_{CO}$ and $X_{CH4}$ from MOPITT (and GOSAT) show good agreement with Pearson's correlation of 0.96 (and 0.97) and mean bias relative to TCCON of -12.81ppb (-7.12 ppb). OCO-2 $X_{CO2}$, on the other hand, has a higher bias and weaker*

*correlation relative to TCCON (correlation of 0.6 and bias of 1.95 ppm) with the least (highest) bias in Pasadena (Manaus).*

**L219:** Wu and Yu, 2018 has been cited but not listed in the References section.

Thank you for pointing this out. We have included this reference.

**Figure 2:** This is just a personal preference - maybe the position of the legends that indicate the name of the species could be fixed.

Thank you for pointing this out. We have included a single legend color-coded by the corresponding species in the top-right panel for clarity and fixed the names of the species. This figure is now moved to the Supplement and renamed as Figure S2 (as a response to the other reviewer's comment).

The CO between ~ 2015 and 2017 is very high for Darwin. Do we know why this is the case?

Thank you for pointing this out. We attribute this peak in CO to Australian seasonal bushfire that occurred between June 2015 to March 2016. A news article is provided here https://www.abc.net.au/news/2016-03-16/satellite-pictures-reveal-scale-of-summers-bushfire-destruction/7232594

**L250:** "The figure shows a clear seasonal cycle in the abundance of CO over all the locations and the seasonal amplitude is higher over Hefei (38.3±0.0 ppb), Sodankyla (37.2±3.9 ppb) and Pasadena (36.0±4.5 ppb) compared to other locations." – Hefei isn't plotted in Figure 2.

Thank you for pointing this out. We wanted to compare the seasonal cycles in Figure 2 with the values of seasonal amplitude in Table 2 in the above text. However, we have removed this part of the text after considering both reviewers' comments. We still retain Table 2 in the main text with the seasonal amplitude values as mentioned.

**L253:** "Furthermore, a large variability in CO is observed in the seasonal amplitude over Burgos (15.5 ppb), Darwin (10.2 ppb), Reunion (9.2 ppb) and Wollongong (8.5 ppb) during this period." – Burgos isn't plotted in Figure 2. Please consider stating "(not shown)" next to "Burgos".

Thank you for this comment. We have removed this part of the text as well (please see previous response).

**L271:** "We also see a decreasing trend in CO in most of the selected TCCON sites (-0.20 to -0.98 % ppb/year), except at Ascension (3.51±0.43 % ppb/year), Pasadena (0.01±0.22 % ppb/year), and Wollongong (0.27±0.35 % ppb/year)." – Why is the trend different over these three TCCON sites?

Thank you for this comment. The significant positive trend estimated at Ascension can be attributed to the transport of biomass emissions from southern Africa and agrees with previous studies that observed a positive trend in space-borne column CO in southern Africa and its vicinity over the ocean (where Ascension is situated) (Buchholz et al., 2021), as well as an increase in burned area and fire activity (Andela et al., 2017). We have revised the text to include this and remove the mention of the positive trends observed at Pasadena and Wollongong as they have high variability compared to the near-negligible trend estimates. It now reads as:

*The site in Ascension is in a small island with virtually no influence from local sources, but it captures the long-range transport of emissions from Africa (Geibel et al., 2010; Feist et al., 2014, Swap et al., 1996). A significant positive trend in $X_{CO}$ is observed in Ascension (3.51±0.43 % ppb/year) with negative $X_{CO}$ trends in the other sites (Table 2). This can be attributed to increase in burned area and transport from southern Africa reported in previous studies (Buchholz et al., 2021; Andela et al., 2017, Borsdorff et al., 2018). This in combination with low trend observed in $X_{CO2}$ over Ascension may be attributed to a decrease in sources (reduced respiration, increase in lower quality fossil-fuels) or an increase in sinks (enhanced photosynthesis) over the African region. Hickman et al., 2021 reported an increasing trend in CO over north equatorial Africa due to decline in biomass burning emissions from a woodier biome.*

Andela, N., Morton, D. C., Giglio, L., Chen, Y., van der Werf, G. R., Kasibhatla, P. S., DeFries, R. S., Collatz, G. J., Hantson, S., Kloster, S., Bachelet, D., Forrest, M., Lasslop, G., Li, F., Mangeon, S., Melton, J. R., Yue, C., and Randerson, J. T.: A human-driven decline in global burned area, Science, 356, 1356–1362, https://doi.org/10.1126/science.aal4108, 2017.

Buchholz, R. R., Worden, H. M., Park, M., Francis, G., Deeter, M. N., Edwards, D. P., Emmons, L. K., Gaubert, B., Gille, J., Martínez-Alonso, S., Tang, W., Kumar, R., Drummond, J. R., Clerbaux, C., George, M., Coheur, P.-F., Hurtmans, D., Bowman, K. W., Luo, M., Payne, V. H., Worden, J. R., Chin, M., Levy, R. C., Warner, J., Wei, Z., and Kulawik, S. S.: Air pollution trends measured from Terra: CO and AOD over industrial, fire-prone, and background regions, Remote Sensing of Environment, 256, 112275, https://doi.org/10.1016/j.rse.2020.112275, 2021.

**L277:** Does Anmyeondo show the highest CO2 trend and Ascension the lowest? If so, please consider stating this explicitly.

Thank you for pointing this out. Since we re-organized this section (as a response to reviewers' comments), we now mentioned this in Section 2.1. We modified the text to reflect the change in adjectives as follows,

*The trend in $CO_2$ is highest over Anmyeondo (0.81± 0.10 % ppm/year), and lowest over Ascension, (0.60±0.01 % ppm/year).*

**Technical Comments:**

**L116:** "all three species" instead of "all these 3 species"

Thank you for the suggestion. We have revised the text accordingly.

**L123:** "during 2012 to 2019" instead of "during the period 2012 to 2019"

Thank you for the suggestion. We have revised the text accordingly.

**L146:** "than its western counterpart" instead of "than its the western counterpart"

Thank you for pointing this out. We have revised the text accordingly.

**L246:** "Firstly, it is informative to understand…" – Please consider changing "informative" to "pertinent".

Thank you for the suggestion. We have removed the text here after considering both the reviewers' comments.

**L264:** "as" instead of "like that is"

Thank you for the suggestion. We have revised the text accordingly.

**L276:** Please consider rephrasing "$CO_2$ and $CH_4$ are showing an increasing trend in all locations" to "Increased $CO_2$ and $CH_4$ trends are observed at all locations".

Thank you for the suggestion. We have revised the text and placed it in section 2.1.

**L278:** For Sodankyla, "high" instead of "higher" and "low" instead of "lower".

Thank you for the suggestion. We have removed discussions on this in the text.

**L279:** "This is possibly due to", not "This is may due to".

Thank you for the suggestion. We have revised the text accordingly and placed it in section 2.1.

**L281:** over "the" Atlantic Ocean

Thank you for the suggestion. We have removed this text after considering both the reviewers' comments.

**L287:** "necessary" instead of "needed".

Thank you for the suggestion. We have removed this text after considering both the reviewers' comments.

**L296:** "… we show in Figure 3" instead of "… we also show in Figure 3"

Thank you for the suggestion. We have revised the text accordingly and renamed Figure 3 as Figure 2.

**L297:** "We provide estimates" instead of "We also provide estimates".

Thank you for the suggestion. We have revised the text accordingly.

**L298:** "… for the period listed in Table 2" instead of "… for the whole analysis period as presented in Table 2".

Thank you for the suggestion. We have revised the text accordingly.

**L300:** "suggest" instead of "suggests".

Thank you for pointing this out. We have revised the text accordingly.

**L321:** "sites in SH" instead of "sites in SH locations".

Thank you for pointing this out. We have revised the text accordingly.

**L352:** "The local enhancement ratio appears to dominate over the regional ratios for Pasadena in all seasons and relative to the regional ratio, the magnitude of local enhancement ratios in CO/CO2 and CO/CH4 are more significant during Fall." – Please consider splitting the sentence into two.

Thank you for the suggestion. We have revised the text accordingly. It now reads as follows,

> *The local enhancement ratio appears to dominate over the regional ratios for Pasadena across all seasons. The local enhancement ratios in $X_{CO}/X_{CO2}$ and $X_{CO}/X_{CH4}$ compared to the regional ratios are more significant during Fall season (SON).*

**L410:** "comes from" instead of "is coming from"

Thank you for the suggestion. We have revised the text accordingly.

**RC2**: 'Comment on egusphere-2024-705', Anonymous Referee #2, 03 Jun 2024

Mottungan et al., analyses the variations between CO, CO2, and CH4 data from the ground-based TCCON network. They use different regression techniques and enhancement ratio algorithms to understand and separate the regional and local sources at the studied TCCON sites. In general, the paper is scientifically sound; however, it is still a subject to few modifications that need to be addressed before publishing, as outlined below.

*We thank the reviewer for the valuable comments that have greatly improved this manuscript. We have addressed the major and minor comments (black) below with our response in blue.*

**General comments:**

Can the authors clarify if method 2 refers to the calculation of enhancement ratios in previous studies (i.e., Line 115)? If yes please say that directly in the Methods section, if not then an additional analysis to these methods would be very useful. We need to understand better how these methods compare to other generally used methods in the science community, as well as uncertainties between methods.

*Thank you for the comment. We revised the text to now refer to the enhancement ratios calculated using all three methods (1,2,3). The details of these methods are outlined in Section 2.2 of Data and Methods. Based on other reviewer's comments as well, the paragraph now reads as follows:*

> *The enhancement ratio between species X and Y is calculated by mainly two methods: the first is from a linearly regressed slope of X and Y (Andreae et al., 1988a, 1988b) and the second is by dividing the excess of X by the excess of Y (Andreae and Merlet, 2001) (See Methods 1 and 2 in Sect. 2.2 respectively). The first approach of enhancement ratio estimation using regression slopes is difficult to infer when emitted or locally produced species mix with different air masses (e.g., advection from the nearby sources or mixed air masses) downwind of the dominant source where measurements are made. This is especially the case for vertically integrated quantities like the column measurements (either ground-, aircraft- and satellite-based) (Cheng et al., 2017; Halliday et al., 2019; Tang et al., 2019) where vertical information of the species abundance is practically absent. If the emission or plume concentration is significantly larger than the background, the ratio from the regression slope approach does not change (Brigg et al., 2016). But, when emission of the species mixes with different 'backgrounds' than a relatively uniform field, the abundances of X and Y change due to mixing and/or photochemical loss (Mauzerall et al., 1998; Yokelson et al., 2013; Guyon et al., 2005); thus, making it difficult to track the locally emitted contribution to the observed abundance. The latter approach of using excess of the species requires a proper understanding of the background concentration to derive the excess abundance along with the instantaneous concentration of the species, which is not available in most cases. Vertical and horizontal transport also complicates the interpretation of abundance and assessment of local and regional source influences at a particular location (Chatfield et al., 2020). A combination of these two approaches have also been used in previous studies (Hedelius et al., 2018) (Method 3 in Sect. 2.2). Here, we utilize the column measurements of CO, $CO_2$, and $CH_4$ (denoted as $X_{CO}$, $X_{CO2}$, $X_{CH4}$) from the Total Carbon Column Observing*

*Network (TCCON) (Wunch et al., 2011) to understand these variations in the column abundances."*

Please, use XCO2, XCO and XCH4 in the whole paper when referring to column measurements, instead of CO2, CO and CH4.

Thank you for the suggestion. We have revised the text and figures and used $X_{CO}$, $X_{CO2}$ and $X_{CH4}$ in the manuscript wherever we mention the column abundance of these species.

Table 3, Figure 6, Table 4 and 5 do we have any uncertainty estimates or error bars?

Thank you for pointing this out. We have updated Table 3, Figure 6, Table 4, and Table 5 with uncertainty estimates (standard deviations).

**Specific comments:**

Line 81: could you please change 'emission/enhancement' to 'emission or enhancement' and comment on the difference between the two.

Thank you for the suggestion. We have revised the text and added the following comment before this line,

> *Emission (enhancement) ratios are defined as ratios of excess abundance across two species, often in units of mass flux (molar) when the concentrations of the species are estimated near (away from) the emission source (Andreae, 2019; Lefer et al., 1994).*

Line 91: although this is a known method, a reference or two might be useful

Thank you for the suggestion. We have added references when introducing the methods in the Introduction section.

Line 110: this sentence is slightly confusing. 'because of emissions': emission are not transported or transformed in the atmosphere, the atmospheric amounts that are a result of emissions are transported. Perhaps re-phrase.

Thank you for pointing this out. We have revised the text as follows,

> *Specifically, we introduce a combination of established local and bulk regression algorithms in deriving enhancement ratios of the column abundances between these three species to understand their relationships **because these constituents are being mixed, dispersed, transported, and transformed in the atmosphere.***

Line 126: Please see https://amt.copernicus.org/articles/10/2209/2017/ for the original convention of writing XCO2, CO2 should be a subscript.

Thank you for the suggestion. We have updated the text and figures and used $X_{CO2}$, $X_{CO}$, and $X_{CH4}$ when column abundances of these species are mentioned.

Line 127 'in the global carbon' -> 'in global carbon'. Note, I won't be editing all the text for grammar errors or better readability, unless there is a larger error. There are a few grammar errors in the text, nothing major but please re-read the text carefully.

Thank you for pointing this out. We have revised the text to correct any grammatical errors like this.

Section 2.1 Please explain why are these sites selected and not others (i.e., Lauder)?

Thank you for this comment. Our choice of these 11 sites was based on the following,

1. Long-term overlap between 2012 and 2019 as well as satellite data.
2. Representativeness of processes at each location (fires, biogenic, combustion, mixed combustion, and regional transport)

Although we agree that there are limitations in choosing these sites and there are other sites that can meet these criteria, these 11 sites were used to study the multi-species enhancement ratios and their co-variability as a proof of concept to understand their common sources and underlying processes. We have added these limitations in the Conclusion section. i.e.,

*We do, however, want to note that additional work is needed for a more robust estimation of the BERr and the regional enhancement, considering the high variability observed across the species in BERr (as in Figure 3) which consequently leads to higher variability in the regional enhancement estimates."*

*Including additional sites and a longer time period from the newer software version of the TCCON data (GGG2020) will also aid in reducing the uncertainty of the regional versus local enhancements, and generalizability of inferred source/transport signatures."*

Section 2.1 Why May 2018? Is there another, better date when we also have TCCON measurements at other sites?

Thank you for the suggestion. We have revised the text and replaced Figure 1 to show an average across 2012-2019 (only 2015 – 2019 for OCO-2) for the satellite data and the TCCON sites.

Line 137: 'Qualitatively' Could the authors point to the numbers that suggest this?

Thank you for pointing this out. We have revised the text as shown below and quantified the agreement between the satellites and TCCON measurements using Pearson's correlation and mean bias (satellite – TCCON). In response to the other reviewer's comments, this now reads as:

*$X_{CO}$ and $X_{CH4}$ from MOPITT (and GOSAT) show good agreement with Pearson's correlation of 0.96 (and 0.97) and mean bias relative to TCCON of -12.81 ppb (-7.12 ppb). OCO-2 $X_{CO2}$, on the other hand, has a higher bias and weaker correlation relative to TCCON (correlation of 0.6 and bias of 1.95 ppm) with the least (highest) bias in Pasadena (Manaus).*

Figure 1: These are all column measurement so should use XCO2 etc.

Thank you for the comment. We have updated Figure 1 and used this notation to denote column measurements.

Table 1: This could go into an Appendix/Supplement

Line 175: 'The data period and a summary of the characteristic' - The table shows the data period and reference; however, the characteristics are not shown. I suggested moving Table 1 to an Appendix; however, if the authors keep the Table I would suggest to summarize the region/source influence and meteorology characteristic described in this section in Table 1. For this publication, it is more important than the Data Period information. Data Period can go to Appendix or Supplement.

Thank you for this comment. We have added a column on "Source Features" in Table 1 for the sites based on the spatial variability of the emission sources. We have included this table in the main text to explicitly provide details to the reader about the sites used for this study.

Line 227: what is the explanation for taking the difference of average morning values from the average afternoon values?

Thank you for this comment. As mentioned in Yokelson et al. 2013, the species abundances are especially high within the boundary layer during morning hours before sufficient mixing till afternoon. These high values can thus skew the enhancement ratios. The boundary layer influence is considered as the background influence in Method 2 and is removed to calculate the excess in abundance. Wunch et al., 2009 also used this approach to remove any spectroscopic and solar-zenith angle-related errors when calculating daily anomalies.

We also added a sentence in the text to reflect this:

*The difference between morning and afternoon values of the abundance minimizes 1) the influence of high concentration of the species within the boundary layer in the morning (Yokelson et al., 2013); and 2) spectroscopic errors (Wunch et al., 2009).*

Wunch, D., Wennberg, P. O., Toon, G. C., Keppel-Aleks, G., and Yavin, Y. G.: Emissions of greenhouse gases from a North American megacity, Geophys. Res. Lett, 36, https://doi.org/10.1029/2009GL039825, 2009.

Yokelson, R. J., Andreae, M. O., & Akagi, S. K.: Pitfalls with the use of enhancement ratios or normalized excess mixing ratios measured in plumes to characterize pollution sources and aging, Atmos. Meas. Tech., 6(8), 2155–2158, https://doi.org/10.5194/amt-6-2155-2013, 2013.

I am not sure if section 3.1 (including Figure 2 and Table 2) is needed in the main text. These sites were all already analysed in other publications and the conclusions of this section agree with those publications. Instead, I would move this to the Supplement and only copy the important analysis when discussing the enhancement ratios in later Sections (e.g. line 322). Or perhaps mention the most importance points/publications when discussing each site in Section 2.1. I would however keep Figure 3 and the analysis and focus this section on that.

Thank you for the suggestion. We have moved Figure 2 to the Supplement (as Figure S2), and revised Section 3.1 to discuss about Figure 3 (now Figure 2) and the co-variation of the three species. The discussions about the temporal variability as before in Section 3.1 are added in Section 2.1 when discussing the sites.

Figure 4: Explain the error bars, how was it calculated?

For the bulk enhancement ratio (BERr) in Method 1, the error bar is the weighted error calculated from the associated errors of the regression slopes (BERr) estimated using three regression algorithms (Ordinary Least Squares, Geometric Mean, and York).

For AERa in Method 2, the error bar is the weighted error calculated from the standard deviations of the AERa's derived by 1) removing the boundary layer influence and 2) by removing the $5^{th}$ percentile.

For AERr in Method 3, the error bar is the weighted error calculated from the associated errors of the regression slopes (AERr) using the three regression algorithms used in Method 1. We added this sentence to the text:

> *The regression slopes are calculated using three algorithms as in Method 1 on the anomalies calculated from Method 2. AERr is the weighted average of the regression slopes and their associated errors similar to Eq. (7).*

We also added Equations 7 and 9 to reflect these: i.e.,

$$BERr = \left(\frac{BERr_{OLS}}{\sigma^2_{OLS}} + \frac{BERr_{GM}}{\sigma^2_{GM}} + \frac{BERr_{York}}{\sigma^2_{York}}\right) \left(\frac{\sigma^2_{OLS}\sigma^2_{GM}\sigma^2_{York}}{\sigma^2_{OLS}\sigma^2_{GM}+\sigma^2_{OLS}\sigma^2_{York}+\sigma^2_{GM}\sigma^2_{York}}\right) \qquad (7)$$

> *where BERr is the bulk enhancement ratio (or the weighted average of the slopes calculated from three regression algorithms). The weights are based on the associated errors ($\sigma$) from each regression algorithm.*

> *The average AERa is the weighted average of AERa calculated using the AERa from 1) boundary layer influence and 2) the $5^{th}$ percentile methods. The weights are based on the errors (standard deviations) of $C'_{Spc}$ based on (1) and (2).*

$$AERa = \left(\frac{AERa_1}{\sigma^2_1} + \frac{AERa_2}{\sigma^2_2}\right) \left(\frac{\sigma^2_1\sigma^2_2}{\sigma^2_1+\sigma^2_2}\right) \qquad (9)$$

Lines 364-367: can the authors explain why we see this behaviour? Same question goes for the whole paragraph. Please provide more explanation.

Thank you for this comment. We have revised and mentioned the possible reasons for observing the differences in local and regional enhancements across the seasons as shown below from the text,

> *We see in Figure 4 that the seasonal variation of regional and local enhancement ratios at different measurement sites reveals the presence of seasonally varying driving factors in the bulk enhancement ratios. The local enhancement ratio appears to dominate over the regional ratios for Pasadena across all seasons. The local enhancement ratios in $X_{CO}/X_{CO2}$ and $X_{CO}/X_{CH4}$ compared to the regional ratios are more significant during Fall season (SON).* **This may be due to the poor dependency between transported $CH_4$ or $CO_2$ coming from biospheric sources or any non-combustion sources of CO. This is evident in Figure S3 which shows a significant peak in the abundance of $X_{CO2}$ during Fall months over Pasadena, but not in $X_{CO}$.** *Furthermore, the low value of regional enhancement ratio in $X_{CH4}/X_{CO2}$ during Summer over Pasadena* **may be associated with the poor correlation among the species from independent sources or from biospheric sinks of $CO_2$ (see Tables S1 and S2)**. *Similar seasonal variation is observed at Wollongong where it shows a dominant influence of local enhancements of species ratios for most of the seasons. Relative to the regional ratio, the magnitude of local enhancement ratio in $X_{CH4}/X_{CO2}$ is more significant during the months of DJF, which is the summer season in SH. The seasonal variation of $X_{CO}/X_{CH4}$ follows a different pattern in Wollongong with the regional influence dominating for all seasons except JJA (winter in SH).* **This can be attributed to similar chemical loss of CO and $CH_4$ through OH especially in spring and summer in SH (Lelieveld et al., 2016; Fisher et al., 2015).** *The seasonal variation of species enhancement ratio in $X_{CH4}/X_{CO2}$ and $X_{CO}/X_{CH4}$ at Darwin follows similar variations as that in Wollongong although there are differences in absolute magnitude* **due to seasonal bushfire occurrences and fire emissions across south and north Australia.** *The regional enhancement ratio in $X_{CO}/X_{CO2}$ dominates during DJF (summer) and SON (spring) months at Darwin whereas the local enhancement ratio dominates in other seasons. A large difference of about 10 ppb/ppm is also observed between local and regional enhancement ratio in $X_{CO}/X_{CO2}$ during JJA (winter) months.*

Lines 398-405: This should be in the Method section

Thank you for this suggestion. We have moved these lines on calculating the regional and local enhancements in Section 2.2 (for Methods).

Line 477: This is an interesting comparison, add as a separate subsection.

Thank you for this suggestion. We have revised this part into subsection 3.5.

References: Appreciate all the reference; however, the publication does have a higher than usual number of references. I careful review of the references might be useful.

Thank you for this suggestion. We have revised and reduced the number of references in the manuscript.

**Technical comments:**

Define CO, CO2 and CH4 in the abstract

Thank you for this suggestion. We have added the name of the species in the abstract.

Same goes for the main text, including other species (e.g., OH)

Thank you for this suggestion. We have added the name of such species when it first appears in the text.

Line 64: fossil fuel –> fossil-fuel, be consistent throughout the text

Thank you for pointing this out. We have revised the text accordingly by using "fossil-fuels".

Line 77: ICOS, define since everything else was defined

Thank you for pointing this out. We have added the definition of "Integrated Carbon Observation System" for ICOS in the text.

Line 108: ground based -> ground-based

Thank you for this suggestion. We have revised the text accordingly.

Line 122: 'As mentioned' - Terms like this are not necessary, just start with 'We make'

Thank you for this suggestion. We have revised the text accordingly.

Line 123: period from 2012 to 2019

Thank you for the suggestion. We have revised the text as "during 2012 to 2019".

Line 134: FTS - You don't use this abbreviation, so it is not needed

Thank you for pointing this out. We have removed this abbreviation and revised the text accordingly.

Line 128: data sets -> datasets

Thank you for pointing this out. We have revised the text accordingly.

Line 133: be consistent 11 and six, use either numbers of words, or re-check the journal guideline.

Thank you for the suggestion. We have used "11" and "six" following the manuscript guidelines outlined in the Numbers section of https://www.atmospheric-measurement-techniques.net/submission.html#math which says,

> "For items other than units of time or measure, use words for cardinal numbers less than 10; use numerals for 10 and above (e.g. three flasks, seven trees, 6 m, 9 d, 10 desks)."

Line 148: Sodankyla -> Sodankylä, please, also re-check that original spelling is used for all sites.

Thank you for pointing this out. We have revised the text accordingly to reflect the correct spelling for the sites.

Line 156: 'at measurement locations in' -> 'at' is enough

Thank you for the suggestion. We have revised the text accordingly.

Line 262-264: Sentences like this are not really needed. Just start with the next sentence and reference Figure 2.

Thank you for the suggestion. We have removed this text and most of the discussions in this section based on the previous comments.

Section 3.2: Add these as subsection 3.2 and 3.3 (i.e., Enhancement Ratio as 3.2)

Thank you for this suggestion. We have revised the section headings accordingly.

Line 575: The link doesn't work anymore

Thank you for pointing this out. We have updated the link in the text with a working link https://eccad.sedoo.fr

Line 575: 'for' shows up blue (i.e., as a link)

Thank you for pointing this out. We have removed 'for' in the hyperlink.